# Monotone Missing Data: A Blessing and a Curse

**Santtu Tikka**                                                                    *santtu.tikka@jyu.fi*
*Department of Mathematics and Statistics*
*University of Jyvaskyla, Finland*

**Juha Karvanen**                                                                  *juha.t.karvanen@jyu.fi*
*Department of Mathematics and Statistics*
*University of Jyvaskyla, Finland*

**Reviewed on OpenReview:** *https://openreview.net/forum?id=kVthdlAVks*

## Abstract

Monotone missingness is commonly encountered in practice when a missing measurement compels another measurement to be missing. Because of the simpler missing data pattern, monotone missing data is often viewed as beneficial from the perspective of practical data analysis. However, in graphical missing data models, monotonicity has implications for the identifiability of the full law, i.e., the joint distribution of actual variables and response indicators. In the general nonmonotone case, the full law is known to be nonparametrically identifiable if and only if specific graphical structures are not present. We show that while monotonicity may enable the identification of the full law despite some of these structures, it also prevents the identification in certain cases that are identifiable without monotonicity. The results emphasize the importance of proper treatment of monotone missingness in the analysis of incomplete data.

## 1 Introduction

Missing data is ubiquitous across all fields of scientific study and it has the potential to severely impact the results of statistical analyses. In particular, monotone missingness occurs when a missing measurement implies that another measurement must also be missing. In longitudinal studies, a monotone missing data pattern is encountered when dropout is permanent, meaning that subjects do not return to the study after missing one measurement. For instance, in a clinical trial where response variables $Y_1$, $Y_2$ and $Y_3$ are measured at three consecutive monthly visits, a subject who drops out after the first visit will have both $Y_2$ and $Y_3$ missing. Monotone missing data also arise if there are logical constraints or technical restrictions between the measurements. For instance, if the information on the number of children (of a person) is missing, the information on the children's ages will be missing as well.

The missing data literature has been moving from the classical characterization: *missing completely at random* (MCAR), *missing at random* (MAR) and *missing not at random* (MNAR) (Rubin, 1976; Little & Rubin, 2002) towards more sophisticated assumptions which are often expressed using graphical models (Daniel et al., 2012; Mohan et al., 2013; Mohan & Pearl, 2014b; Karvanen, 2015). One of the major goals related to nonparametric missing data models has been to characterize the set of missing data distributions that are identifiable as functionals of the observed data distribution. Several graphical criteria and identifiability algorithms have been developed for this purpose (e.g., Tian, 2017; Bhattacharya et al., 2020; Mohan & Pearl, 2021; Guo et al., 2023), including a sound and complete graphical criterion by Nabi et al. (2020). As a result of these efforts, it is known when the joint distribution variables and response indicators is identifiable in nonparametric MNAR models with nonmonotone missingness.

In contrast, identifiability under monotone missingness is far less studied despite the prevalence of monotone missing data mechanisms in real-world scenarios. Identification strategies for monotone missingness usually consider MCAR or MAR settings, and identifiability is achieved by imposing identifying restrictions, such as

the complete case missing value restriction (Little, 1993), available case missing value restriction (Molenberghs et al., 1998), neighboring-case missing value restriction (Thijs, 2002), and specific donor-based identification restrictions (Chen & Sadinle, 2019). Identifying restrictions have also been developed for specific MNAR settings (e.g., Kenward et al., 2003; Tang et al., 2003).

Inference under monotone missingness is often viewed as a simpler problem than scenarios involving non-monotone missingness. For instance, Sikov (2018) states that "The advantage of the monotone missingness condition is that it considerably simplifies the analysis of the data." It turns out, that identification under monotone missingness is far from simple and it is not a subset of general nonparametric identification under missing data, but a distinct problem that only partially overlaps the general problem. The presence of monotonic relationships in the missingness mechanism implies that the probability of some combinations of values of the response indicators is exactly zero. This reduces the number of parameters in the missingness mechanism to be identified, leading to new identification results. On the other hand, monotonicity violates the positivity assumptions that are explicitly or implicitly needed in many identification results with nonmonotone missing data rendering them inapplicable under monotone missingness (Nabi et al., 2024).

In this paper, we consider missing data models represented by directed acyclic graphs (DAGs) and scenarios where the assumption of a monotonic relationship between response indicators enables us to identify distributions of interest that would otherwise be nonidentifiable, and the converse, where the same assumption renders otherwise identifiable distributions nonidentifiable. To the best of our knowledge, there are no previous graphical criteria or algorithms for determining identifiability or nonidentifiability of the missing data distribution under monotone missing data in nonparametric MNAR settings for missing data DAGs.

The rest of the paper is organized as follows. Section 2 introduces the notation and the relevant definitions. Section 3 considers missing data models and monotone missingness. Section 4 discusses identifiability in missing data models and the applicability of previous identifiability results for nonmonotone missingness under monotone missingness. Sections 5 and 6 present new results for identifiability and nonidentifiability under monotone missingness, respectively. Section 7 concludes the paper with a discussion.

## 2 Notations and Definitions

We use capital letters to denote random variables or vertices, and small letters to denote the values or value assignments of random variables. We use bold letters to denote sets and vectors of random variables, vertices, or values.

A directed graph $\mathcal{G}$ is a pair $(\mathbf{V}, \mathbf{E})$ where $\mathbf{V}$ is the vertex set and $\mathbf{E}$ is the set of directed edges (i.e., pairs $(V_i, V_j)$, $V_i \neq V_j$, $V_i, V_j \in \mathbf{V}$). An edge from $V_i$ to $V_j$ is also denoted by $V_i \rightarrow V_j$. A directed graph over a set of vertices $\mathbf{V}$ is denoted by $\mathcal{G}(\mathbf{V})$. If the edge $(V_i, V_j)$ exists in $\mathcal{G}$, we say that $V_i$ is a parent of $V_j$ and $V_j$ is a child of $V_i$. The set of parents of a vertex $V_i$ in $\mathcal{G}$ is denoted by $\mathrm{pa}_{\mathcal{G}}(V_i)$ and the set of children is denoted by $\mathrm{ch}_{\mathcal{G}}(V_i)$, respectively. If the graph $\mathcal{G}$ is clearly determined by the context, we will simply write $\mathrm{pa}(V_i)$ and $\mathrm{ch}(V_i)$ for clarity.

A directed path is a sequence of distinct edges $(E_i)_{i=1}^{k}$ such that $E_i = (V_i, V_{i+1})$ and each vertex $V_i$ may have at most one incoming and one outgoing edge in the sequence. A directed path where the first and the last vertex are the same is called a cycle. If a directed path exists in $\mathcal{G}$ from $V_i$ to $V_j$, then $V_i$ is an ancestor of $V_j$ and $V_j$ is a descendant of $V_i$. The set of ancestors of a vertex $V_i$ in $\mathcal{G}$ is denoted by $\mathrm{an}_{\mathcal{G}}(V_i)$ and the set of descendants by $\mathrm{de}_{\mathcal{G}}(V_i)$, respectively (omitting again the graph if the context is clearly defined). A directed acyclic graph (DAG) is a directed graph that contains no cycles.

We consider DAGs whose vertices represent random variables. For simplicity, we will denote both vertices and the associated random variables using the same symbols. A statistical model of a DAG $\mathcal{G}(\mathbf{V})$ is a set of distributions that factorize according to the structure of the DAG as follows

$$p(\mathbf{V}) = \prod_{V_i \in \mathbf{V}} p(V_i \mid \mathrm{pa}_{\mathcal{G}}(V_i)), \tag{1}$$

Whenever a joint distribution is compatible with a DAG, the conditional independence constraints of the distribution can be derived from the DAG using the d-separation criterion (Pearl, 1995; 2009). In general,

we do not make a distinction between d-separation statements (such as $\mathbf{X}$ is d-separated from $\mathbf{Y}$ given $\mathbf{Z}$) from conditional independence statements (such as $\mathbf{X}$ is independent of $\mathbf{Y}$ given $\mathbf{Z}$). However, we will also consider models that contain deterministic relationships between the variables of interest, meaning that d-separation will not imply conditional independence in all instances for such models. Thus, we will only denote conditional independence constraints as $\mathbf{X} \perp\!\!\!\perp \mathbf{Y} \mid \mathbf{Z}$ and explicitly explain when such statements are not implied by the DAG due to deterministic relationships.

## 3 Missing Data Models

Missing data models are sets of distributions over a set of random variables $\mathbf{V}$ where $\mathbf{V}$ can be partitioned into four distinct sets: the set of fully observed variables $\mathbf{O}$, the set of partially observed variables $\mathbf{X}^{(1)}$, the set of observed proxy variables $\mathbf{X}$, and the set of response indicators $\mathbf{R}$ (sometimes referred to as missingness indicators or simply indicators). Each partially observed variable $X^{(1)} \in \mathbf{X}^{(1)}$ has a corresponding observed proxy and a response indicator that have the following deterministic relationship

$$X = \begin{cases} X^{(1)} & \text{if } R_X = 1, \\ \text{NA} & \text{if } R_X = 0, \end{cases} \tag{2}$$

where NA (not available) denotes a missing value. In other words, $R_X = 1$ means that the true value of the variable $X^{(1)}$ was observed, and $R_X = 0$ indicates that it is missing. For a set $\mathbf{Y}^{(1)} \subseteq \mathbf{X}^{(1)}$ of partially observed variables, we denote the corresponding set of response indicators by $R_\mathbf{Y}$ defined as $R_\mathbf{Y} = \cup_{Y_i^{(1)} \in \mathbf{Y}^{(1)}} \{R_{Y_i}\}$.

By (2), we can factorize the joint distribution of $\mathbf{V}$ as

$$p(\mathbf{V}) = p(\mathbf{O}, \mathbf{X}, \mathbf{X}^{(1)}, \mathbf{R}) = p(\mathbf{X}|\mathbf{X}^{(1)}, \mathbf{R})p(\mathbf{O}, \mathbf{X}^{(1)}, \mathbf{R}),$$

where the nondeterministic term $p(\mathbf{O}, \mathbf{X}^{(1)}, \mathbf{R})$ is the *full law* which can be further partitioned into two terms: the *target law* $p(\mathbf{O}, \mathbf{X}^{(1)})$ and the *missingness mechanism* $p(\mathbf{R}|\mathbf{O}, \mathbf{X}^{(1)})$. Finally, the information available under missing data is represented by the *observed data law* $p(\mathbf{O}, \mathbf{X}, \mathbf{R})$. This distribution is the one we actually have access to. We note that the term "missingness mechanism" is sometimes used to refer to the set of response indicators $\mathbf{R}$ or its members instead of their conditional distribution. Similarly, the target law is sometimes simply referred to as the joint distribution (see e.g., Mohan et al., 2013; Mohan & Pearl, 2014a), but to avoid ambiguity, we use the terms "target law" and "full law" to distinguish between the two joint distributions $p(\mathbf{O}, \mathbf{X}^{(1)})$ and $p(\mathbf{O}, \mathbf{X}^{(1)}, \mathbf{R})$, respectively.

Missing data models can be represented by *missing data DAGs* (m-DAGs). A DAG $\mathcal{G}$ is a missing data DAG if it has the following properties: the vertex set of $\mathcal{G}$ is $\mathbf{O} \cup \mathbf{X} \cup \mathbf{X}^{(1)} \cup \mathbf{R}$; for each $X \in \mathbf{X}$, $\text{pa}_\mathcal{G}(X) = \{X^{(1)}, R_X\}$ and $\text{ch}_\mathcal{G}(X) = \emptyset$; and for each $R_X \in \mathbf{R}$, $\deg_\mathcal{G}(R_X) \cap (\mathbf{O} \cup \mathbf{X}^{(1)}) = \emptyset$. A missing data model associated with a missing data DAG $\mathcal{G}$ is the set of joint distributions $p(\mathbf{O}, \mathbf{X}, \mathbf{X}^{(1)}, \mathbf{R})$ that factorize as

$$p(\mathbf{O}, \mathbf{X}, \mathbf{X}^{(1)}, \mathbf{R}) = \prod_{X \in \mathbf{X}} p(X|R_X, X^{(1)}) \prod_{V \in \mathbf{O} \cup \mathbf{X}^{(1)} \cup \mathbf{R}} p(V \mid \text{pa}_\mathcal{G}(V)). \tag{3}$$

Conditional independence constraints of $p(\mathbf{O}, \mathbf{X}^{(1)}, \mathbf{R})$ can be determined via d-separation in m-DAGs analogously to DAGs.

The missingness mechanism of a missing data model may contain monotonic relationships between response indicators, meaning that missingness in one variable always renders another variable to be missing as well. We define this property as follows.

**Definition 1.** *The missingness mechanism of a missing data model associated with an m-DAG $\mathcal{G}(\mathbf{V})$ is locally monotone with respect to $(R_Z, R_W)$ if the edge $R_Z \rightarrow R_W$ exists in $\mathcal{G}$ and $p(\mathbf{R} = \mathbf{r}|\mathbf{O}, \mathbf{X}^{(1)}) = 0$ for all value assignments $\mathbf{r}$ to $\mathbf{R}$ where $r_Z = 0$ and $r_W = 1$. Furthermore, we say that such a value assignment $\mathbf{r}$ to $\mathbf{R}$ violates monotonicity.*

As a shorthand notation, we will denote the assumption that the missingness mechanism is locally monotone with respect to $(R_Z, R_W)$ as $R_Z \geq R_W$. Graphically, we denote the same assumption as $R_Z \xrightarrow{\geq} R_W$ or

$R_Z \xrightarrow{\geq} R_W$. Furthermore, we say that a missing data model is monotone if the missingness mechanism is locally monotone with respect to at least one pair of response indicators.

We note that the concept of monotone missingness is used to refer to different properties of the missingness mechanism in literature. For example, Cui et al. (2017) use the term "monotone missing data mechanism" to describe that the missingness probability is a monotonic function of a set of covariates and a response variable. Miao et al. (2016) define a "monotone missing mechanism" in a similar way. In this paper, we will only consider monotonicity in accordance with Definition 1.

## 4 Identifiability in Missing Data Models

In this section, we revisit some identifiability results for nonmonotone missingness and discuss their applicability under monotone missingness. We begin by defining the notion of identifiability, which is also referred to as *recoverability* in missing data models (Mohan et al., 2013; Mohan & Pearl, 2014a;b)

**Definition 2.** *Given a missing data model $\mathcal{M}$ associated with an m-DAG $\mathcal{G}(\mathbf{V})$, an estimand or a probabilistic query $Q$ is said to be identifiable if $Q$ can be expressed in terms of the observed data distribution $p(\mathbf{X}, \mathbf{O}, \mathbf{R})$, that is, if $Q_1 = Q_2$ for every pair of distributions $p_1, p_2 \in \mathcal{M}$ such that $p_1(\mathbf{X}, \mathbf{O}, \mathbf{R}) = p_2(\mathbf{X}, \mathbf{O}, \mathbf{R})$.*

The query of interest $Q$ for nonparametric identification in missing data models is typically either the target law $p(\mathbf{O}, \mathbf{X}^{(1)})$, the full law $p(\mathbf{O}, \mathbf{X}^{(1)}, \mathbf{R})$, or some functional of them, but other quantities such as causal effects can also be considered (Shpitser et al., 2015). Because the component $\prod_{X \in \mathbf{X}} p(X|R_X, X^{(1)})$ in (3) is deterministic, we can ignore it in all identifiability considerations. For the same reason, we also omit the observed proxy variables and the corresponding deterministic edges related to them from all figures.

Typically, definitions of identifiability also include a positivity assumption such as $p(\mathbf{X} = \mathbf{x}, \mathbf{O} = \mathbf{o}, \mathbf{R} = \mathbf{r}) > 0$ (Tian, 2017), which we omit from our definition because our goal is to consider identifiability not only under nonmonotone missingness but also under monotone missingness, where some events have zero probability. This means that we must consider a different positivity assumptions depending on whether the missingness mechanism is assumed to be locally monotone or not. Thus, for the remainder of the paper, we will assume that $p(\mathbf{X} = \mathbf{x}, \mathbf{O} = \mathbf{o}, \mathbf{R} = \mathbf{r}) > 0$ for all value assignments in scenarios where monotonicity is not assumed, and for the scenarios where monotonic relationships between response indicators are present, we assume that $p(\mathbf{X} = \mathbf{x}, \mathbf{O} = \mathbf{o}, \mathbf{R} = \mathbf{r}) > 0$ for only those value assignments $\mathbf{r}$ to $\mathbf{R}$ that do not violate monotonicity. In simpler terms, we will assume a positive probability for events whose probability is not zero due to deterministic relationships between response indicators.

The missingness mechanism $p(\mathbf{R}|\mathbf{O}, \mathbf{X}^{(1)})$ is a key component for the identification of both the target law and the full law due to the following identities:

$$p(\mathbf{O}, \mathbf{X}^{(1)}) = \frac{p(\mathbf{O}, \mathbf{X}^{(1)}, \mathbf{R} = 1)}{p(\mathbf{R} = 1|\mathbf{O}, \mathbf{X}^{(1)})},$$

$$p(\mathbf{O}, \mathbf{X}^{(1)}, \mathbf{R}) = \frac{p(\mathbf{O}, \mathbf{X}^{(1)}, \mathbf{R} = 1)}{p(\mathbf{R} = 1|\mathbf{O}, \mathbf{X}^{(1)})} p(\mathbf{R}|\mathbf{O}, \mathbf{X}^{(1)}),$$

where $\mathbf{R} = 1$ means that all response indicators have a value assignment of 1. In other words, identifiability of the target law is equivalent to the identifiability of $p(\mathbf{R} = 1|\mathbf{O}, \mathbf{X}^{(1)})$ and identifiability of the full law is equivalent to the identifiability of the missingness mechanism as the numerator $p(\mathbf{O}, \mathbf{X}^{(1)}, \mathbf{R} = 1)$ is always identified from the observed data distribution. This is why methods for identification in missing data models often target the missingness mechanism instead of the full law or target law directly. These identities also hold for locally monotone missingness mechanisms under the corresponding positivity assumption.

There are two important graphical structures related to nonidentifiability in missing data DAGs. The first is a *self-censoring edge* where a partially observed variable is a parent of its corresponding response indicator, i.e., $X^{(1)} \to R_X$ (also referred to as "self-masking", e.g., Mohan et al., 2018). The second is a *collider* where a partially observed variable and its response indicators are the parents of a response indicator of another partially observed variable, i.e., $X^{(1)} \to R_Y \leftarrow R_X$. Self-censoring edges and colliders are the only structures

that render the full law nonparametrically nonidentifiable in missing data DAGs (Bhattacharya et al., 2020; Nabi et al., 2020). Self-censoring also renders the target law non-identifiable (Mohan et al., 2013). We say that a response indicator $R_Y$ is *colluded* if a collider structure $X^{(1)} \to R_Y \leftarrow R_X$ is present in the m-DAG.

Building on the seminal works of Mohan et al. (2013) and Mohan & Pearl (2014a;b), Nabi et al. (2020) provided a sound and complete criterion for full law identifiability under a general nonparametric setting without assumptions of monotonicity. Importantly, when no colliders or self-censoring edges are present, this criterion provides an identifying functional that relies on the odds ratio (OR) parameterization of the missingness mechanism (Yun Chen, 2006). Denote $\mathbf{R}_{-k} = \mathbf{R} \setminus R_k$, $\mathbf{R}_{\prec k} = \{R_1, \ldots, R_{k-1}\}$, and $\mathbf{R}_{\succ k} = \{R_{k+1}, \ldots R_K\}$. Now we can write

$$p(\mathbf{R} \mid \mathbf{O}, \mathbf{X}^{(1)}) = \frac{1}{Z} \prod_{k=1}^{K} p(R_k \mid \mathbf{R}_{-k} = 1, \mathbf{O}, \mathbf{X}^{(1)}) \prod_{k=2}^{K} \mathrm{OR}(R_k, \mathbf{R}_{\prec k} \mid \mathbf{R}_{\succ k} = 1, \mathbf{O}, \mathbf{X}^{(1)}), \tag{4}$$

where

$$\mathrm{OR}(R_k, \mathbf{R}_{\prec k} \mid \mathbf{R}_{\succ k} = 1, \mathbf{O}, \mathbf{X}^{(1)}) = \frac{p(R_k \mid \mathbf{R}_{\succ k} = 1, \mathbf{R}_{\prec k}, \mathbf{O}, \mathbf{X}^{(1)})}{p(R_k = 1 \mid \mathbf{R}_{\succ k} = 1, \mathbf{R}_{\prec k}, \mathbf{O}, \mathbf{X}^{(1)})} \frac{p(R_k = 1 \mid \mathbf{R}_{-k} = 1, \mathbf{O}, \mathbf{X}^{(1)})}{p(R_k \mid \mathbf{R}_{-k} = 1, \mathbf{O}, \mathbf{X}^{(1)})},$$

and $Z$ is the normalizing constant. However, this identifying functional is no longer valid if the missingness mechanism is locally monotone as this implies that some of the terms in the denominator of the OR terms will be zero. This will occur even if we restrict our attention to value assignments that do not violate monotonicity, because the term $p(R_k = 1 \mid \mathbf{R}_{\succ k} = 1, \mathbf{R}_{\prec k}, \mathbf{O}, \mathbf{X}^{(1)})$ will be zero for some $k$ as $R_k$ is conditioned on all other response indicators, and the normalizing term evaluates this probability for all value assignments of $\mathbf{R}_{\prec k}$. In contrast, an earlier identifiability result by Bhattacharya et al. (2020) based on propensity scores can be applied under monotone missingness, as it only involves terms related to response indicators of the form $p(R_k = 1 \mid \mathrm{pa}_{\mathcal{G}}(R_k))\big|_{(\mathrm{pa}_{\mathcal{G}}(R_k) \cap \mathbf{R}) = 1}$ in a denominator term, meaning that only nonzero probabilities (assuming positivity) are considered in the denominator of the identifying functional.

There are also methods for target law identification that can be applied under monotone missingness. The propensity score approach of Bhattacharya et al. (2020) is equally valid for target law identification. The MID algorithm by Shpitser et al. (2015) also remains applicable under monotone missingness as it attempts to identify $p(R_k = 1 \mid \mathrm{pa}_{\mathcal{G}}(R_k))\big|_{(\mathrm{pa}_{\mathcal{G}}(R_k) \cap \mathbf{R}) = 1}$ for each response indicator $R_k$. A graphical criterion by Mohan et al. (2013) states that if no partially observed variable is an ancestor of its own response indicator, then the target law is identifiable, and its expression is

$$p(\mathbf{X}^{(1)}, \mathbf{O}) = \prod_{V_i \in \mathbf{X}^{(1)} \cup \mathbf{O}} p(V_i \mid \mathrm{pa}_{\mathcal{G}}(V_i) \cap (\mathbf{X}^{(1)} \cup \mathbf{O}))$$

The conditional independence restrictions implied by the non-ancestrality assumption also hold if monotonicity is assumed.

Outside of graphical missing data models, the impact of functional relationships on identification has been studied in the context of causal inference by Chen & Darwiche (2024). In their approach, some variables are assumed functionally dependent (i.e., deterministic) on their parents and subsequently removed via functional elimination. Unfortunately, this approach does not directly generalize to the setting of full law identifiability and monotone missingness, as monotonicity does not impose a true deterministic relationship between response indicators.

While identification strategies such as the propensity score based method of Bhattacharya et al. (2020) or the MID algorithm of Shpitser et al. (2015) remain valid under monotone missingness, their scope is simultaneously limited by the standard positivity assumption. Intuitively, because $p(\mathbf{X} = \mathbf{x}, \mathbf{O} = \mathbf{o}, \mathbf{R} = \mathbf{r}) = 0$ for values assignments that violate positivity, the missingness mechanism has fewer parameters (and consequently the full law), and thus identification is easier in a sense when the missingness model is monotone. Importantly, the crucial collider structure does not always prohibit identification under monotone missingness. In the following sections, we will focus on these special instances where monotonicity makes otherwise nonidentifiable quantities identifiable, and the converse, where monotonicity makes otherwise identifiable quantities nonidentifiable.

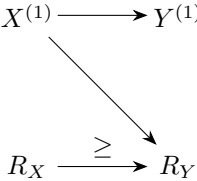

Figure 1: An example m-DAG where the monotonicity assumption enables the full law to be identified.

## 5 Identifiability Gained under Monotonicity

A locally monotone missingness mechanism can enable us to identify the full law in scenarios where identifiability could not be achieved otherwise. As an example, we consider the graph of Figure 1 where the monotonicity assumption allows us to identify the full law. The full law is not identifiable without this assumption because of the colluder $X^{(1)} \to R_Y \leftarrow R_X$ (Bhattacharya et al., 2020).

As a practical example of Figure 1, consider an intervention program to improve the physical condition of the participants. Variable $X^{(1)}$ stands for the result of a physical test at the beginning of the intervention and $Y^{(1)}$ represents the result of the same test after the intervention. The pre-interventional physical condition $X^{(1)}$ affects both the post-interventional condition $Y^{(1)}$ and the participant's decision $R_Y$ to complete the intervention and the final test. The monotonicity $R_X \geq R_Y$ occurs because the measurement of $X^{(1)}$ is a prerequisite for participating in the intervention.

We begin by considering the identifiability of the full law under the assumption of monotonicity, meaning that that $R_X \geq R_Y$. This means that

$$p(X^{(1)}, Y^{(1)}, R_X = 0, R_Y = 1) = 0$$

from which we obtain that

$$p(X^{(1)}, Y^{(1)}, R_X = 0, R_Y = 0) = p(X^{(1)}, Y^{(1)}, R_X = 0)$$

and consequently

$$
\begin{aligned}
&p(X^{(1)}, Y^{(1)}, R_X = 0, R_Y = 0) \\
&= p(X^{(1)}, Y^{(1)}, R_X = 0) \\
&= p(Y^{(1)} \,|\, X^{(1)}, R_X = 0) p(X^{(1)} \,|\, R_X = 0) p(R_X = 0) \\
&= p(Y^{(1)} \,|\, X^{(1)}, R_X = 1, R_Y = 1) p(X^{(1)} \,|\, R_X = 1) p(R_X = 0) \\
&= p(Y \,|\, X, R_X = 1, R_Y = 1) p(X \,|\, R_X = 1) p(R_X = 0)
\end{aligned}
$$

where we used the following facts

$$
\begin{aligned}
& X^{(1)} \perp\!\!\!\perp R_X, \\
& Y^{(1)} \perp\!\!\!\perp R_X \,|\, X^{(1)}, \\
& Y^{(1)} \perp\!\!\!\perp \{R_Y, R_X\} \,|\, X^{(1)}.
\end{aligned}
$$

Thus $p(X^{(1)}, Y^{(1)}, R_X = 0, R_Y = 0)$ is identifiable. The term $p(X^{(1)}, Y^{(1)}, R_X = 1, R_Y = 1)$ is directly identifiable from the complete cases. It remains to show that $p(X^{(1)}, Y^{(1)}, R_X = 1, R_Y = 0)$ is also identifiable. To show this, we can write

$$
\begin{aligned}
&p(X^{(1)}, Y^{(1)}, R_X = 1, R_Y = 0) \\
&= p(Y^{(1)} \,|\, X^{(1)}, R_X = 1, R_Y = 0) p(X^{(1)}, R_X = 1, R_Y = 0) \\
&= p(Y^{(1)} \,|\, X^{(1)}, R_X = 1, R_Y = 1) p(X^{(1)}, R_X = 1, R_Y = 0) \\
&= p(Y \,|\, X, R_X = 1, R_Y = 1) p(X, R_X = 1, R_Y = 0)
\end{aligned}
$$

where we again used the fact that $Y^{(1)} \perp\!\!\!\perp \{R_X, R_Y\} \,|\, X^{(1)}$. Thus $p(X^{(1)}, Y^{(1)}, R_X = 1, R_Y = 0)$ is identifiable. By combining the above cases, we conclude that the full law $p(X^{(1)}, Y^{(1)}, R_X, R_Y)$ is identifiable in the m-DAG of Figure 1 under the assumption of monotonicity.

It is evident that we cannot leverage monotonicity when self-censoring edges are present for identifiability purposes. However, as the previous example shows, monotonicity can be beneficial when colluders are present. We present several generalizations of the previous example in the form of graphical criteria and show how identifiability can be regained when the m-DAG contains colluders. We defer all proofs in Sections 5 and 6 to Appendices A and B, respectively. As a starting point, the following definition characterizes multiple simultaneous colluders affecting the same response indicator.

**Definition 3.** *A pair $(\mathbf{C}^{(1)}, R_Y)$ is a maximal colluder in an m-DAG $\mathcal{G}$ if $\mathcal{G}$ contains the edges $C_i^{(1)} \to R_Y$ and $R_{C_i} \to R_Y$ for all $C_i^{(1)} \in \mathbf{C}^{(1)}$ and there does not exist $Z^{(1)} \in \mathbf{X}^{(1)} \setminus \mathbf{C}^{(1)}$ such that $\mathcal{G}$ contains the edges $Z^{(1)} \to R_Y$ and $R_Z \to R_Y$.*

For convenience, we use the notation $\min R_{\mathbf{C}}$ to denote the random variable that takes the smallest value among members of $R_{\mathbf{C}}$. In other words, $\min R_{\mathbf{C}}$ is 0 if at least one response indicator in $R_{\mathbf{C}}$ is 0 and otherwise its value is 1, i.e., when all response indicators in $R_{\mathbf{C}}$ have the value 1. The following result is immediate.

**Theorem 1.** *Let $\mathcal{G}$ be an m-DAG with a maximal colluder $(\mathbf{C}^{(1)}, R_Y)$. If $\min R_{\mathbf{C}} \geq R_Y$ then $p(R_Y \,|\, \mathrm{pa}_{\mathcal{G}}(R_Y))\big|_{\min R_{\mathbf{C}}=0}$, i.e., $p(R_Y \,|\, \mathrm{pa}_{\mathcal{G}}(R_Y))$ under the value assignment $\min R_{\mathbf{C}} = 0$, is identifiable, and*

$$p(R_Y = 1 \,|\, \mathrm{pa}_{\mathcal{G}}(R_Y))\big|_{\min R_{\mathbf{C}}=0} = 0,$$
$$p(R_Y = 0 \,|\, \mathrm{pa}_{\mathcal{G}}(R_Y))\big|_{\min R_{\mathbf{C}}=0} = 1.$$

Theorem 1 essentially states that we can always identify the conditional distribution of $R_Y$ for those value assignments of the relevant response indicators where the monotonicity is violated. Thus it remains to consider value assignments that do not violate monotonicity, i.e., the case with $R_{\mathbf{C}} = 1$.

When $R_Y$ has parents that include partially observed variables whose response indicators are not parents of $R_Y$, we can identify the conditional distribution of $R_Y$ if $R_Y$ is conditionally independent of the respective response indicators of the partially observed variables that are parents of $R_Y$.

**Theorem 2.** *Let $\mathcal{G}$ be an m-DAG with a maximal colluder $(\mathbf{C}^{(1)}, R_Y)$. If $\min R_{\mathbf{C}} \geq R_Y$ and*

$$R_Y \perp\!\!\!\perp \mathbf{R}' \,|\, \mathrm{pa}_{\mathcal{G}}(R_Y),$$

*where $\mathbf{R}' = R_{\mathrm{pa}_{\mathcal{G}}(R_Y) \cap \mathbf{X}^{(1)}} \setminus \mathrm{pa}_{\mathcal{G}}(R_Y)$, then $p(R_Y \,|\, \mathrm{pa}_{\mathcal{G}}(R_Y))\big|_{R_{\mathbf{C}}=1}$ is identifiable, and*

$$p(R_Y \,|\, \mathrm{pa}_{\mathcal{G}}(R_Y))\big|_{R_{\mathbf{C}}=1} = p(R_Y \,|\, \mathrm{pa}_{\mathcal{G}}(R_Y), \mathbf{R}')\big|_{R_{\mathbf{C}}=1, \mathbf{R}'=1}$$

In other words, $\mathbf{R}'$ is a set of response indicators for those partially observed variables that are parents of $R_Y$ but that are not themselves parents of $R_Y$.

Figure 2a illustrates a scenario considered by Theorem 2. In this m-DAG $(X^{(1)}, R_Y)$ is a maximal colluder such that $R_X \geq R_Y$, and we have that $R_Y \perp\!\!\!\perp \mathbf{R}' \,|\, \mathrm{pa}_{\mathcal{G}}(R_Y)$ which in this case translates to $R_Y \perp\!\!\!\perp R_Z \,|\, X^{(1)}, Z^{(1)}, R_X$. Now, we can write

$$p(R_Y | X^{(1)}, Z^{(1)}, R_X = 1) = p(R_Y | X^{(1)}, Z^{(1)}, R_X = 1, R_Z = 1),$$

where the right-hand side is identifiable from the observed data distribution.

Another way to identify the conditional distribution of $R_Y$ is to instead identify the conditional distribution of the other partially observed variables that are parents of $R_Y$ given $R_Y$ and the other parents of $R_Y$, but whose response indicators are not parents of $R_Y$.

**Theorem 3.** *Let $\mathcal{G}$ be an m-DAG with a maximal colluder $(\mathbf{C}^{(1)}, R_Y)$. If $\min R_{\mathbf{C}} \geq R_Y$ and*

$$\mathbf{Z} \perp\!\!\!\perp R_{\mathbf{Z}} \,|\, R_Y \cup (\mathrm{pa}_{\mathcal{G}}(R_Y) \setminus \mathbf{Z})$$

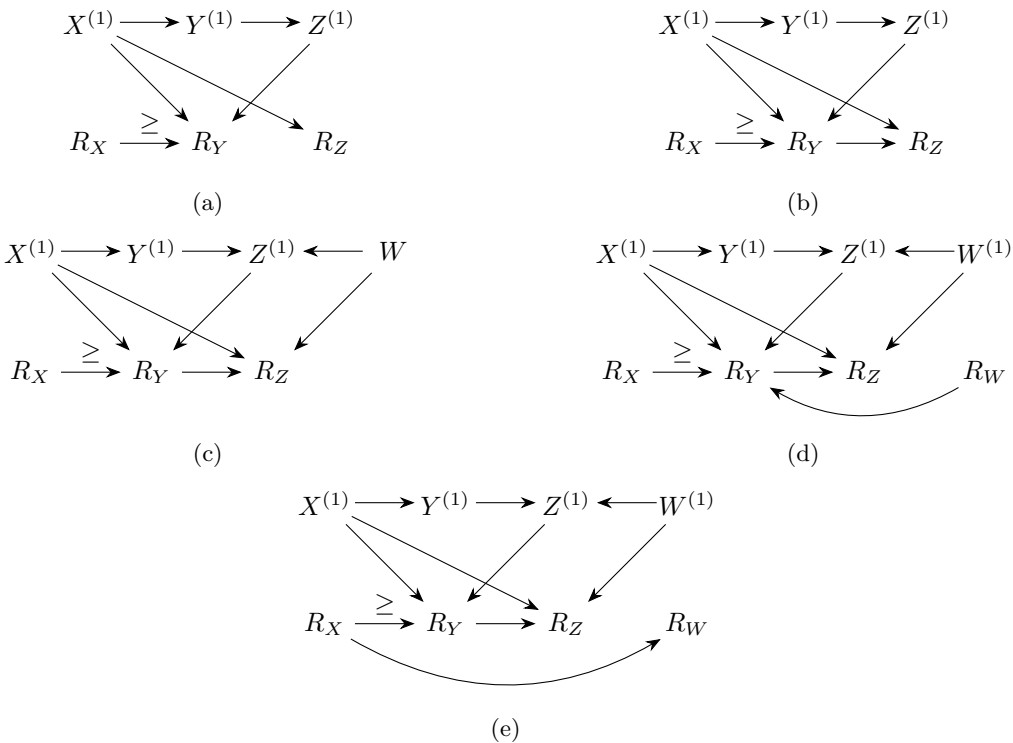

Figure 2: Example m-DAGs for Theorems 2–5 where the conditional distribution of $R_Y$ is identifiable when the missingness mechanism is locally monotone with respect to $(R_X, R_Y)$ but not otherwise.

*where $\mathbf{Z}^{(1)} = \{Z^{(1)} \in \mathbf{X}^{(1)} \mid Z^{(1)} \in \mathrm{pa}_{\mathcal{G}}(R_Y), R_Z \notin \mathrm{pa}_{\mathcal{G}}(R_Y)\}$, then $p(R_Y \mid \mathrm{pa}_{\mathcal{G}}(R_Y))\big|_{R_{\mathbf{C}}=1}$ is identifiable, and*

$$p(R_Y \mid \mathrm{pa}_{\mathcal{G}}(R_Y))\big|_{R_{\mathbf{C}}=1}$$
$$= \frac{p(\mathbf{Z}^{(1)} \mid R_{\mathbf{Z}}, R_Y, \mathrm{pa}_{\mathcal{G}}(R_Y) \setminus \mathbf{Z}^{(1)})p(R_Y, \mathrm{pa}_{\mathcal{G}}(R_Y) \setminus \mathbf{Z}^{(1)})}{\sum_{R_Y} p(\mathbf{Z}^{(1)} \mid R_{\mathbf{Z}}, R_Y, \mathrm{pa}_{\mathcal{G}}(R_Y) \setminus \mathbf{Z}^{(1)})p(R_Y, \mathrm{pa}_{\mathcal{G}}(R_Y) \setminus \mathbf{Z}^{(1)})}\Bigg|_{R_{\mathbf{C}}=1, R_{\mathbf{Z}}=1}.$$

An example use case of Theorem 3 is shown in Figure 2b. In contrast to Figure 2a, there is now an additional edge from $R_Y$ to $R_Z$, which means that we can no longer use Theorem 2 to identify the conditional distribution of $R_Y$. Again, $(X^{(1)}, R_Y)$ is a maximal collider such that $R_X \geq R_Y$, but now we have that $\mathbf{Z}^{(1)} \perp\!\!\!\perp R_{\mathbf{Z}} \mid R_Y \cup (\mathrm{pa}_{\mathcal{G}}(R_Y) \setminus \mathbf{Z}^{(1)})$ which reads as $Z^{(1)} \perp\!\!\!\perp R_Z \mid R_X, R_Y, X^{(1)}$ in this instance. By Theorem 3, we can write

$$p(R_Y|X^{(1)}, Z^{(1)}, R_X = 1)$$
$$= \frac{p(Z^{(1)}|R_Z = 1, R_Y, R_X = 1, X^{(1)})p(R_Y, R_X = 1, X^{(1)})}{\sum_{R_Y} p(Z^{(1)}|R_Z = 1, R_Y, R_X = 1, X^{(1)})p(R_Y, R_X = 1, X^{(1)})}.$$

Another strategy is to identify the conditional distribution of $R_Y$ is analogous to the previous but also employs a fully observed proxy variable.

**Theorem 4.** *Let $\mathcal{G}$ be an m-DAG with a maximal collider $(\mathbf{C}^{(1)}, R_Y)$. If $\min R_{\mathbf{C}} \geq R_Y$ and there exists $\mathbf{W} \subset (\mathbf{O} \setminus \mathrm{pa}_{\mathcal{G}}(R_Y))$ such that*

$$\mathbf{Z}^{(1)} \perp\!\!\!\perp R_{\mathbf{Z}} \mid \mathbf{W} \cup R_Y \cup (\mathrm{pa}_{\mathcal{G}}(R_Y) \setminus \mathbf{Z}^{(1)})$$

*where* $\mathbf{Z}^{(1)} = \{Z^{(1)} \in \mathbf{X}^{(1)} \mid Z^{(1)} \in \mathrm{pa}_{\mathcal{G}}(R_Y), R_Z \notin \mathrm{pa}_{\mathcal{G}}(R_Y)\}$, *then* $p(R_Y \mid \mathrm{pa}_{\mathcal{G}}(R_Y))\big|_{R_{\mathbf{C}}=1}$ *is identifiable,*
*and*

$$p(R_Y \mid \mathrm{pa}_{\mathcal{G}}(R_Y))\big|_{R_{\mathbf{C}}=1}$$

$$= \frac{\sum_{\mathbf{W}} p(\mathbf{Z}^{(1)} \mid \mathbf{W}, R_Y, \mathrm{pa}_{\mathcal{G}}(R_Y) \setminus \mathbf{Z}^{(1)}, R_{\mathbf{Z}}) p(\mathbf{W}, R_Y, \mathrm{pa}_{\mathcal{G}}(R_Y) \setminus \mathbf{Z}^{(1)})}{\sum_{\mathbf{W}, R_Y} p(\mathbf{Z}^{(1)} \mid \mathbf{W}, R_Y, \mathrm{pa}_{\mathcal{G}}(R_Y) \setminus \mathbf{Z}^{(1)}, R_{\mathbf{Z}}) p(\mathbf{W}, R_Y, \mathrm{pa}_{\mathcal{G}}(R_Y) \setminus \mathbf{Z}^{(1)})}\Bigg|_{R_{\mathbf{C}}=1, R_{\mathbf{Z}}=1}.$$

Figure 2c depicts a scenario where Theorems 2 and 3 cannot be applied to identify the conditional distribution of $R_Y$, but Theorem 4 applies. The required conditional independence $\mathbf{Z}^{(1)} \perp\!\!\!\perp R_{\mathbf{Z}} \mid \mathbf{W} \cup R_Y \cup (\mathrm{pa}_{\mathcal{G}}(R_Y) \setminus \mathbf{Z}^{(1)})$ corresponds to $Z^{(1)} \perp\!\!\!\perp R_Z \mid W, X^{(1)}, R_X, R_Y$, which holds in this m-DAG. We can write

$$p(Z^{(1)}, X^{(1)}, R_Y, R_X = 1)$$
$$= \sum_W p(Z^{(1)} \mid R_Z = 1, R_Y, R_X = 1, X^{(1)}, W) p(X^{(1)}, W, R_X = 1, R_Y),$$

and we have that

$$p(R_Y \mid X^{(1)}, Z^{(1)}, R_X = 1)$$
$$= \frac{\sum_W p(Z^{(1)} \mid R_Z = 1, R_Y, R_X = 1, X^{(1)}, W) p(X^{(1)}, W, R_X = 1, R_Y)}{\sum_{R_Y, W} p(Z^{(1)} \mid R_Z = 1, R_Y, R_X = 1, X^{(1)}, W) p(X^{(1)}, W, R_X = 1, R_Y)}.$$

If a fully observed proxy variable required by Theorem 4 does not exist, it may still be possible to use a partially observed variable instead. However, in this case we must also be able to identify the conditional distribution of this partially observed variable, thus we employ another assumption of conditional independence, as outlined by the next theorem.

**Theorem 5.** *Let $\mathcal{G}$ be an m-DAG with a maximal colluder $(\mathbf{C}^{(1)}, R_Y)$. If $\min R_{\mathbf{C}} \geq R_Y$ and there exists $\mathbf{W}^{(1)} \subset (\mathbf{X}^{(1)} \setminus \mathrm{pa}_{\mathcal{G}}(R_Y))$ such that*

$$\mathbf{Z}^{(1)} \perp\!\!\!\perp R_{\mathbf{Z}} \cup (R_{\mathbf{W}} \setminus \mathrm{pa}_{\mathcal{G}}(R_Y)) \mid \mathbf{W}^{(1)} \cup R_Y \cup (\mathrm{pa}_{\mathcal{G}}(R_Y) \setminus \mathbf{Z}^{(1)}), and$$
$$\mathbf{W}^{(1)} \perp\!\!\!\perp R_{\mathbf{W}} \setminus \mathrm{pa}_{\mathcal{G}}(R_Y) \mid \{R_Y\} \cup (\mathrm{pa}_{\mathcal{G}}(R_Y) \setminus \mathbf{Z}^{(1)}),$$

*where $\mathbf{Z} = \{Z^{(1)} \in \mathbf{X}^{(1)} \mid Z^{(1)} \in \mathrm{pa}_{\mathcal{G}}(R_Y), R_Z \notin \mathrm{pa}_{\mathcal{G}}(R_Y)\}$, then $p(R_Y \mid \mathrm{pa}_{\mathcal{G}}(R_Y))\big|_{R_{\mathbf{C}}=1}$ is identifiable, and*

$$p(R_Y \mid \mathrm{pa}_{\mathcal{G}}(R_Y))\big|_{R_{\mathbf{C}}=1} = \frac{\sum_{\mathbf{W}^{(1)}} Q}{\sum_{\mathbf{W}^{(1)}, R_Y} Q}\Bigg|_{R_{\mathbf{C}}=1, R_{\mathbf{Z}}=1, R_{\mathbf{W}}=1},$$

*where*

$$Q = p(\mathbf{Z}^{(1)} \mid \mathbf{W}^{(1)}, R_{\mathbf{Z}}, R_{\mathbf{W}}, R_Y, \mathrm{pa}_{\mathcal{G}}(R_Y) \setminus (\mathbf{Z}^{(1)} \cup R_{\mathbf{W}}))$$
$$\times p(\mathbf{W}^{(1)} \mid R_{\mathbf{W}}, R_Y, \mathrm{pa}_{\mathcal{G}}(R_Y) \setminus (\mathbf{Z}^{(1)} \cup R_{\mathbf{W}})) p(R_Y, \mathrm{pa}_{\mathcal{G}}(R_Y) \setminus \mathbf{Z}^{(1)})$$

As the conditions of Theorem 5 are rather complicated, we will illustrate the theorem using two examples where $\mathbf{W}^{(1)}$ is a singleton $\{W^{(1)}\}$. The first example considers the case where $R_W \in \mathrm{pa}_{\mathcal{G}}(R_Y)$ and the corresponding m-DAG is shown in Figure 2d. In this scenario, the first required conditional independence is $Z^{(1)} \perp\!\!\!\perp R_Z \mid W^{(1)}, X^{(1)}, R_X, R_W, R_Y$ which holds in the m-DAG, and the second condition is trivially satisfied. This allows us to write

$$p(Z^{(1)}, X^{(1)}, R_Y, R_W = 1, R_X = 1)$$
$$= \sum_{W^{(1)}} p(Z^{(1)} \mid R_Y, R_Z = 1, R_W = 1, R_X = 1, X^{(1)}, W^{(1)}) p(R_Y, W^{(1)}, X^{(1)}, R_X = 1, R_W = 1),$$

and we have that

$$p(R_Y \mid X^{(1)}, Z^{(1)}, R_W = 1, R_X = 1)$$
$$= \frac{\sum\limits_{W^{(1)}} p(Z^{(1)} \mid R_Y, R_Z = 1, R_W = 1, R_X = 1, X^{(1)}, W^{(1)}) p(R_Y, W^{(1)}, X^{(1)}, R_X = 1, R_W = 1)}{\sum\limits_{R_Y, W^{(1)}} p(Z^{(1)} \mid R_Y, R_Z = 1, R_W = 1, R_X = 1, X^{(1)}, W^{(1)}) p(R_Y, W^{(1)}, X^{(1)}, R_X = 1, R_W = 1)}.$$

Intuitively, Theorems 4 and 5 operate almost identically when $R_{\mathbf{W}} \subset \mathrm{pa}_{\mathcal{G}}(R_Y)$, because the partially observed variables $\mathbf{W}^{(1)}$ essentially act as an observed proxy in Theorem 5 in such cases. In the second example scenario depicted in Figure 2e, we have that $R_W \notin \mathrm{pa}_{\mathcal{G}}(R_Y)$. Now, the required conditional independence restrictions are

$$Z^{(1)} \perp\!\!\!\perp \{R_Z, R_W\} \,|\, W^{(1)}, X^{(1)}, R_X, R_Y,$$
$$W^{(1)} \perp\!\!\!\perp R_W \,|\, X^{(1)}, R_X, R_Y,$$

which hold in the m-DAG. We can write

$$p(Z^{(1)}, X^{(1)}, R_Y, R_X = 1)$$
$$= \sum_{W^{(1)}} p(Z^{(1)}|R_Y, R_Z = 1, R_W = 1, R_X = 1, X^{(1)}, W^{(1)})$$
$$\times p(W^{(1)}|R_Y, X^{(1)}, R_X = 1, R_W = 1)p(R_Y, X^{(1)}, R_X = 1)$$

and thus obtain $p(R_Y|X^{(1)}, Z^{(1)}, R_W = 1, R_X = 1)$ as in the first example.

It is important to keep in mind the functional relationships between response indicators when considering d-separation statements under monotone missing data. For example, in Figure 3a it is the case that $Y^{(1)} \perp\!\!\!\perp R_Y \,|\, R_X$ if there is no monotonic relationship between $R_X$ and $R_Y$ irrespective of the values of $R_X$ and $R_Y$. However, if the monotonic relationship is present, the conditional independence only holds only if $R_X = 1$ because only then is $R_Y$ a true random variable. We note that in Theorems 2–5, we only use the conditional independence statements implied by d-separation under the value assignment of $\min R_{\mathbf{C}} = 1$, thus avoiding false implications of conditional independence due to the functional relationships arising from monotone missingness. However, if there are other monotonic relationships present in the missing data model than those related to the colluder structure, then it may be the case that the required conditional independence properties no longer hold even if implied by d-separation due to the functional relationships induced by monotonicity. One might also consider D-separation (Geiger et al., 1990) that takes into account deterministic variables. However, D-separation may also lead to false conclusions about conditional independence under monotonicity, as monotonicity only enforces determinism for a subset of value assignments of the response indicators.

As a tool for full law identification, Theorems 1–5 should be applied to identify the conditional distributions of response indicators that are colluded. If all conditional distributions of colluded response indicators can be identified in this way, we can attempt to identify the remaining conditional distributions of response indicators using existing methods (e.g., Bhattacharya et al., 2020).

## 6    Identifiability Lost under Monotonicity

Monotonicity is not always beneficial for identification tasks in missing data models. As an example, we consider the m-DAG of Figure 3a. Without assumptions of monotonicity, the full law is identifiable as the m-DAG does not contain self-censoring edges or colluders. The identifying formula can be derived as follows:

$$p(X^{(1)}, Y^{(1)}, R_X, R_Y)$$
$$= p(X^{(1)}|Y^{(1)}, R_X, R_Y)p(Y^{(1)}|R_X, R_Y)p(R_X, R_Y)$$
$$= p(X^{(1)}|Y^{(1)}, R_X = 1, R_Y = 1)p(Y^{(1)}|R_X, R_Y = 1)p(R_X, R_Y)$$
$$= p(X|Y, R_X = 1, R_Y = 1)p(Y|R_X, R_Y = 1)p(R_X, R_Y) \tag{5}$$

where we used the facts that $X^{(1)} \perp\!\!\!\perp \{R_X, R_Y\} \,|\, Y^{(1)}$ and $Y^{(1)} \perp\!\!\!\perp R_Y \,|\, R_X$. However, if monotonicity is assumed, i.e. $R_X \geq R_Y$, the response indicators $R_X$ and $R_Y$ become functionally dependent, and the conditional independence $Y^{(1)} \perp\!\!\!\perp R_Y \,|\, R_X$ that is critical for obtaining the identifying functional no longer holds. The nonidentifiability construction provided in Appendix C shows that neither the full law nor even the marginal distributions $p(X^{(1)})$ and $p(Y^{(1)})$ are identifiable under the monotonicity constraint. This example has been considered previously by e.g., Mohan (2022).

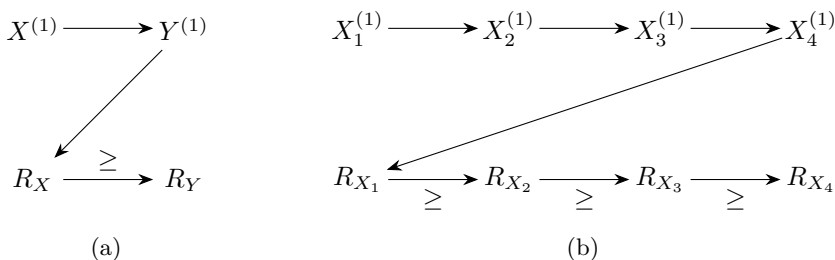

Figure 3: Example m-DAGs where the monotonicity assumption ($R_X \geq R_Y$ in (a) and $R_{X_1} \geq R_{X_2} \geq R_{X_3} \geq R_{X_4}$ in (b)) renders the full law nonidentifiable.

As a practical example of Figure 3a, consider an epidemiological study on hobbies and cognitive abilities in elderly people. At the first stage of the study, the participants are asked to fill out a questionnaire on their hobbies and activities during the past ten years (variable $X^{(1)}$). Those who returned the questionnaire are then invited to participate in measurements of cognitive abilities (variable $Y^{(1)}$). Restricting these measurements to those who returned the questionnaire implies the monotonic relationship $R_X \geq R_Y$. It is reasonable assume that past hobbies and activities have an effect on the current cognitive abilities (edge $X^{(1)} \to Y^{(1)}$). Cognitive abilities may affect the ability and motivation to fill out the questionnaire (edge $Y^{(1)} \to R_X$) but past hobbies and activities do not have a direct effect on the filling of the questionnaire (the absence of self-censoring for $X^{(1)}$). Neither past hobbies and activities nor cognitive abilities have a direct effect on whether cognitive abilities are measured or not because this part of the study is not self-administrated (absence of edges $X^{(1)} \to R_Y$ and $Y^{(1)} \to R_Y$).

The monotonicity assumption $R_X \geq R_Y$ means that the cognitive abilities can be measured only from those individuals who returned the questionnaire. This breaks identifiability because the observed data do not provide information on the relation of cognitive abilities $Y^{(1)}$ and the decision to return the questionnaire $R_X$. Regarding (5), this means that we are not able to identify $p(Y^{(1)}|R_X = 0, R_Y = 1)$ from $p(X, Y, R_X, R_Y)$.

It is well known that self-censoring edges, i.e., edges of the form $X^{(1)} \to R_X$ render the corresponding marginal distribution $p(X^{(1)})$ nonidentifiable (Mohan et al., 2013). Intuitively, monotonicity can induce analogous structures which we call *self-censoring paths* in an m-DAG due to the functional dependency between the response indicators, i.e., a path from a partially observed variable to its own response indicator via other response indicators, all of which have a monotonic relationship. An example of a self-censoring path is presented in Figure 3b. A self-censoring path renders the marginal distribution of the corresponding partially observed variable nonidentifiable under monotonicity (and consequently the target law and the full law). This notion is formalized by the following theorem.

**Theorem 6.** *Let $\mathcal{G}$ be an m-DAG that contains the edge $X_k^{(1)} \to R_{X_1}$ and the edges $R_{X_{j-1}} \to R_{X_j}$ for all $j = 2, \ldots, k$ (a self-censoring path). If $R_{X_{j-1}} \geq R_{X_j}$ for all $j = 2, \ldots, k$, then $p(X_k^{(1)})$ is not identifiable.*

The following corollary is immediate.

**Corollary 1.** *If an m-DAG $\mathcal{G}$ contains a self-censoring path where the response indicators have a monotonic relationship, then neither the full law nor the target law is identifiable.*

In addition, if there is a true dependency between the partially observed variables whose response indicators are part of the self-censoring path, it may not be possible to identify even the marginal distributions of these variables, as demonstrated in Appendix C.

## 7 Discussion

We considered monotone missing data and its implications on nonparametric full law identifiability from a graphical modeling perspective. Specifically, we showed that the colluder structure is not always a detriment to identification under missing data, and conversely, how the self-censoring path structure emerges as a

barrier to identification, analogously to self-censoring. These findings emphasize that monotone missing data must be properly accounted for both in identification and estimation, and not merely dismissed as a special instance of missing data.

In the present work, we focused on missing data models associated with DAGs. In other words, we assumed that no hidden variables are present. Missing data models with hidden variables are often represented via acyclic directed mixed graphs (ADMG) where bidirected edges are used to denote the effects of hidden confounders (Richardson et al., 2023). We note however, that our results directly generalize to such models because Theorems 1–5 only make use of d-separation properties of m-DAGs to derive conditional independence properties, which can be accomplished analogously via m-separation in missing data ADMGs (under value assignments of response indicators where monotonicity is not violated). Similarly, the construction used to prove Theorem 6 is also valid for missing data ADMGs.

The presented work leaves some avenues for potential future research. It may be possible to devise an identifiability algorithm or a graphical criterion for monotone missingness similar to those for nonmonotone missingness. However, this approach would face several challenges. First, the deterministic relationships implied by a locally monotone missingness mechanism make it impractical to leverage d-separation for conditional independence constraints. Furthermore, the deterministic relationships are dependent on the particular value assignment to the response indicators. Second, while the result by Nabi et al. (2020) provides a complete characterization of full law identifiability under nonmonotone missingness, the result relies on the OR factorization in (4) meaning that it does not directly translate to monotone settings. Similarly, there is currently no complete characterization of target law identifiability even under nonmonotone missingness.

The question remains whether the self-censoring path is the only new structure under monotone missing data that renders the full law (and the target law) nonparametrically nonidentifiable. We hypothesize that this is the case. It also seems that monotonicity is only beneficial for full law identification and not for target law identification. Intuitively, this would seem to be true as identification of the target law does not necessitate identification of the entire missingness mechanism, but only the fully observed portion where all response indicators have a value assignment of one, and monotonicity does not have an impact on positivity.

## Acknowledgement

This work was supported by the Research Council of Finland under grant number 368935.

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

# A    Proofs for Section 5 (Identifiability Gained under Monotonicity)

**Theorem 1.** *Let $\mathcal{G}$ be an m-DAG with a maximal colluder $(\mathbf{C}^{(1)}, R_Y)$. If $\min R_{\mathbf{C}} \geq R_Y$ then $p(R_Y \mid \mathrm{pa}_{\mathcal{G}}(R_Y))\big|_{\min R_{\mathbf{C}}=0}$, i.e., $p(R_Y \mid \mathrm{pa}_{\mathcal{G}}(R_Y))$ under the value assignment $\min R_{\mathbf{C}} = 0$, is identifiable, and*

$$p(R_Y = 1 \mid \mathrm{pa}_{\mathcal{G}}(R_Y))\big|_{\min R_{\mathbf{C}}=0} = 0,$$
$$p(R_Y = 0 \mid \mathrm{pa}_{\mathcal{G}}(R_Y))\big|_{\min R_{\mathbf{C}}=0} = 1.$$

*Proof.* The claim follows directly from Definition 1, as the missingness mechanism is locally monotone with respect to each pair $(R_C, R_Y)$, $C^{(1)} \in \mathbf{C}^{(1)}$. $\qquad\square$

**Theorem 2.** *Let $\mathcal{G}$ be an m-DAG with a maximal colluder $(\mathbf{C}^{(1)}, R_Y)$. If $\min R_{\mathbf{C}} \geq R_Y$ and*

$$R_Y \perp\!\!\!\perp \mathbf{R}' \mid \mathrm{pa}_{\mathcal{G}}(R_Y),$$

*where $\mathbf{R}' = R_{\mathrm{pa}_{\mathcal{G}}(R_Y) \cap \mathbf{X}^{(1)}} \setminus \mathrm{pa}_{\mathcal{G}}(R_Y)$, then $p(R_Y \mid \mathrm{pa}_{\mathcal{G}}(R_Y))\big|_{R_{\mathbf{C}}=1}$ is identifiable, and*

$$p(R_Y \mid \mathrm{pa}_{\mathcal{G}}(R_Y))\big|_{R_{\mathbf{C}}=1} = p(R_Y \mid \mathrm{pa}_{\mathcal{G}}(R_Y), \mathbf{R}')\big|_{R_{\mathbf{C}}=1, \mathbf{R}'=1}$$

*Proof.* If $\mathrm{pa}_{\mathcal{G}}(R_Y) \cap \mathbf{X}^{(1)} = \emptyset$, the claim is immediate because $\mathrm{pa}_{\mathcal{G}}(R_Y)$ is fully observed. Suppose now that $\mathrm{pa}_{\mathcal{G}}(R_Y) \cap \mathbf{X}^{(1)} \neq \emptyset$. Then, by the assumed conditional independence, we have that

$$p(R_Y \mid \mathrm{pa}_{\mathcal{G}}(R_Y))\big|_{R_{\mathbf{C}}=1} = p(R_Y \mid \mathrm{pa}_{\mathcal{G}}(R_Y), \mathbf{R}')\big|_{R_{\mathbf{C}}=1, \mathbf{R}'=1},$$

where the right-hand side is a function of the observed data distribution. $\qquad\square$

**Theorem 3.** *Let $\mathcal{G}$ be an m-DAG with a maximal colluder $(\mathbf{C}^{(1)}, R_Y)$. If $\min R_{\mathbf{C}} \geq R_Y$ and*

$$\mathbf{Z} \perp\!\!\!\perp R_{\mathbf{Z}} \mid R_Y \cup (\mathrm{pa}_{\mathcal{G}}(R_Y) \setminus \mathbf{Z})$$

*where $\mathbf{Z}^{(1)} = \{Z^{(1)} \in \mathbf{X}^{(1)} \mid Z^{(1)} \in \mathrm{pa}_{\mathcal{G}}(R_Y), R_Z \notin \mathrm{pa}_{\mathcal{G}}(R_Y)\}$, then $p(R_Y \mid \mathrm{pa}_{\mathcal{G}}(R_Y))\big|_{R_{\mathbf{C}}=1}$ is identifiable, and*

$$p(R_Y \mid \mathrm{pa}_{\mathcal{G}}(R_Y))\big|_{R_{\mathbf{C}}=1}$$
$$= \frac{p(\mathbf{Z}^{(1)} \mid R_{\mathbf{Z}}, R_Y, \mathrm{pa}_{\mathcal{G}}(R_Y) \setminus \mathbf{Z}^{(1)}) p(R_Y, \mathrm{pa}_{\mathcal{G}}(R_Y) \setminus \mathbf{Z}^{(1)})}{\sum_{R_Y} p(\mathbf{Z}^{(1)} \mid R_{\mathbf{Z}}, R_Y, \mathrm{pa}_{\mathcal{G}}(R_Y) \setminus \mathbf{Z}^{(1)}) p(R_Y, \mathrm{pa}_{\mathcal{G}}(R_Y) \setminus \mathbf{Z}^{(1)})}\Bigg|_{R_{\mathbf{C}}=1, R_{\mathbf{Z}}=1}.$$

*Proof.* If $\mathbf{Z}^{(1)} = \emptyset$, the claim follows by observing that $p(R_Y \mid \mathrm{pa}_{\mathcal{G}}(R_Y))\big|_{R_{\mathbf{C}}=1}$ is directly identifiable from the observed data distribution, as the right-hand side may only contain observed proxies, response indicators, and partially observed variables whose response indicators are also parents of $R_Y$.

Suppose now that $\mathbf{Z}^{(1)} \neq \emptyset$. Then, by the assumed conditional independence, we have that

$$p(\mathbf{Z}^{(1)}, R_Y, \mathrm{pa}_{\mathcal{G}}(R_Y) \setminus \mathbf{Z}^{(1)})\big|_{R_{\mathbf{C}}=1}$$
$$= p(\mathbf{Z}^{(1)} \mid R_Y, \mathrm{pa}_{\mathcal{G}}(R_Y) \setminus \mathbf{Z}^{(1)}) p(R_Y, \mathrm{pa}_{\mathcal{G}}(R_Y) \setminus \mathbf{Z}^{(1)})\big|_{R_{\mathbf{C}}=1}$$
$$= p(\mathbf{Z}^{(1)} \mid R_{\mathbf{Z}}, R_Y, \mathrm{pa}_{\mathcal{G}}(R_Y) \setminus \mathbf{Z}^{(1)}) p(R_Y, \mathrm{pa}_{\mathcal{G}}(R_Y) \setminus \mathbf{Z}^{(1)})\big|_{R_{\mathbf{C}}=1, R_{\mathbf{Z}}=1}$$

where both terms on the last line are identifiable from the observed data distribution. Thus $p(\mathbf{Z}^{(1)}, R_Y, \mathrm{pa}_{\mathcal{G}}(R_Y) \setminus \mathbf{Z}^{(1)})\big|_{R_{\mathbf{C}}=1}$ is identifiable, and we can write

$$
\begin{aligned}
&p(R_Y \mid \mathrm{pa}_{\mathcal{G}}(R_Y))\big|_{R_{\mathbf{C}}=1} \\
&= \left. \frac{p(R_Y, \mathrm{pa}_{\mathcal{G}}(R_Y))}{\sum_{R_Y} p(R_Y, \mathrm{pa}_{\mathcal{G}}(R_Y))} \right|_{R_{\mathbf{C}}=1} \\
&= \left. \frac{p(\mathbf{Z}^{(1)}, R_Y, \mathrm{pa}_{\mathcal{G}}(R_Y) \setminus \mathbf{Z}^{(1)})}{\sum_{R_Y} p(\mathbf{Z}^{(1)}, R_Y, \mathrm{pa}_{\mathcal{G}}(R_Y) \setminus \mathbf{Z}^{(1)})} \right|_{R_{\mathbf{C}}=1} \\
&= \left. \frac{p(\mathbf{Z}^{(1)} \mid R_{\mathbf{Z}}, R_Y, \mathrm{pa}_{\mathcal{G}}(R_Y) \setminus \mathbf{Z}^{(1)}) p(R_Y, \mathrm{pa}_{\mathcal{G}}(R_Y) \setminus \mathbf{Z}^{(1)})}{\sum_{R_Y} p(\mathbf{Z}^{(1)} \mid R_{\mathbf{Z}}, R_Y, \mathrm{pa}_{\mathcal{G}}(R_Y) \setminus \mathbf{Z}^{(1)}) p(R_Y, \mathrm{pa}_{\mathcal{G}}(R_Y) \setminus \mathbf{Z}^{(1)})} \right|_{R_{\mathbf{C}}=1, R_{\mathbf{Z}}=1},
\end{aligned}
$$

and the claim follows. Note that despite the monotonicity $\min R_{\mathbf{C}} \geq R_Y$, the denominator in the identifying functional is always positive because of the value assignment $R_{\mathbf{C}} = 1$. $\qquad\square$

**Theorem 4.** *Let $\mathcal{G}$ be an m-DAG with a maximal colluder $(\mathbf{C}^{(1)}, R_Y)$. If $\min R_{\mathbf{C}} \geq R_Y$ and there exists $\mathbf{W} \subset (\mathbf{O} \setminus \mathrm{pa}_{\mathcal{G}}(R_Y))$ such that*

$$
\mathbf{Z}^{(1)} \perp\!\!\!\perp R_{\mathbf{Z}} \mid \mathbf{W} \cup R_Y \cup (\mathrm{pa}_{\mathcal{G}}(R_Y) \setminus \mathbf{Z}^{(1)})
$$

*where $\mathbf{Z}^{(1)} = \{Z^{(1)} \in \mathbf{X}^{(1)} \mid Z^{(1)} \in \mathrm{pa}_{\mathcal{G}}(R_Y), R_Z \notin \mathrm{pa}_{\mathcal{G}}(R_Y)\}$, then $p(R_Y \mid \mathrm{pa}_{\mathcal{G}}(R_Y))\big|_{R_{\mathbf{C}}=1}$ is identifiable, and*

$$
\begin{aligned}
&p(R_Y \mid \mathrm{pa}_{\mathcal{G}}(R_Y))\big|_{R_{\mathbf{C}}=1} \\
&= \left. \frac{\sum_{\mathbf{W}} p(\mathbf{Z}^{(1)} \mid \mathbf{W}, R_Y, \mathrm{pa}_{\mathcal{G}}(R_Y) \setminus \mathbf{Z}^{(1)}, R_{\mathbf{Z}}) p(\mathbf{W}, R_Y, \mathrm{pa}_{\mathcal{G}}(R_Y) \setminus \mathbf{Z}^{(1)})}{\sum_{\mathbf{W}, R_Y} p(\mathbf{Z}^{(1)} \mid \mathbf{W}, R_Y, \mathrm{pa}_{\mathcal{G}}(R_Y) \setminus \mathbf{Z}^{(1)}, R_{\mathbf{Z}}) p(\mathbf{W}, R_Y, \mathrm{pa}_{\mathcal{G}}(R_Y) \setminus \mathbf{Z}^{(1)})} \right|_{R_{\mathbf{C}}=1, R_{\mathbf{Z}}=1}.
\end{aligned}
$$

*Proof.* By the assumed conditional independence, we have that

$$
\begin{aligned}
&p(\mathbf{Z}^{(1)}, R_Y, \mathrm{pa}_{\mathcal{G}}(R_Y) \setminus \mathbf{Z}^{(1)})\big|_{R_{\mathbf{C}}=1} \\
&= \sum_{\mathbf{W}} p(\mathbf{Z}^{(1)}, \mathbf{W}, R_Y, \mathrm{pa}_{\mathcal{G}}(R_Y) \setminus \mathbf{Z}^{(1)})\big|_{R_{\mathbf{C}}=1} \\
&= \sum_{\mathbf{W}} p(\mathbf{Z}^{(1)} \mid \mathbf{W}, R_Y, \mathrm{pa}_{\mathcal{G}}(R_Y) \setminus \mathbf{Z}^{(1)}) p(\mathbf{W}, R_Y, \mathrm{pa}_{\mathcal{G}}(R_Y) \setminus \mathbf{Z}^{(1)})\big|_{R_{\mathbf{C}}=1} \\
&= \sum_{\mathbf{W}} p(\mathbf{Z}^{(1)} \mid \mathbf{W}, R_Y, \mathrm{pa}_{\mathcal{G}}(R_Y) \setminus \mathbf{Z}^{(1)}, R_{\mathbf{Z}}) p(\mathbf{W}, R_Y, \mathrm{pa}_{\mathcal{G}}(R_Y) \setminus \mathbf{Z}^{(1)})\big|_{R_{\mathbf{C}}=1, R_{\mathbf{Z}}=1}
\end{aligned}
$$

thus the claim follows by using the same reasoning as in the proof of Theorem 3. $\qquad\square$

**Theorem 5.** *Let $\mathcal{G}$ be an m-DAG with a maximal colluder $(\mathbf{C}^{(1)}, R_Y)$. If $\min R_{\mathbf{C}} \geq R_Y$ and there exists $\mathbf{W}^{(1)} \subset (\mathbf{X}^{(1)} \setminus \mathrm{pa}_{\mathcal{G}}(R_Y))$ such that*

$$
\begin{aligned}
\mathbf{Z}^{(1)} &\perp\!\!\!\perp R_{\mathbf{Z}} \cup (R_{\mathbf{W}} \setminus \mathrm{pa}_{\mathcal{G}}(R_Y)) \mid \mathbf{W}^{(1)} \cup R_Y \cup (\mathrm{pa}_{\mathcal{G}}(R_Y) \setminus \mathbf{Z}^{(1)}), \text{ and} \\
\mathbf{W}^{(1)} &\perp\!\!\!\perp R_{\mathbf{W}} \setminus \mathrm{pa}_{\mathcal{G}}(R_Y) \mid \{R_Y\} \cup (\mathrm{pa}_{\mathcal{G}}(R_Y) \setminus \mathbf{Z}^{(1)}),
\end{aligned}
$$

*where $\mathbf{Z} = \{Z^{(1)} \in \mathbf{X}^{(1)} \mid Z^{(1)} \in \mathrm{pa}_{\mathcal{G}}(R_Y), R_Z \notin \mathrm{pa}_{\mathcal{G}}(R_Y)\}$, then $p(R_Y \mid \mathrm{pa}_{\mathcal{G}}(R_Y))\big|_{R_{\mathbf{C}}=1}$ is identifiable, and*

$$
p(R_Y \mid \mathrm{pa}_{\mathcal{G}}(R_Y))\big|_{R_{\mathbf{C}}=1} = \left. \frac{\sum_{\mathbf{W}^{(1)}} Q}{\sum_{\mathbf{W}^{(1)}, R_Y} Q} \right|_{R_{\mathbf{C}}=1, R_{\mathbf{Z}}=1, R_{\mathbf{W}}=1},
$$

*where*

$$Q = p(\mathbf{Z}^{(1)} \mid \mathbf{W}^{(1)}, R_{\mathbf{Z}}, R_{\mathbf{W}}, R_Y, \mathrm{pa}_{\mathcal{G}}(R_Y) \setminus (\mathbf{Z}^{(1)} \cup R_{\mathbf{W}}))$$
$$\times p(\mathbf{W}^{(1)} \mid R_{\mathbf{W}}, R_Y, \mathrm{pa}_{\mathcal{G}}(R_Y) \setminus (\mathbf{Z}^{(1)} \cup R_{\mathbf{W}})) p(R_Y, \mathrm{pa}_{\mathcal{G}}(R_Y) \setminus \mathbf{Z}^{(1)})$$

*Proof.* Let $\mathbf{D} = \mathbf{C}^{(1)} \cup \{W^{(1)} \in \mathbf{X}^{(1)} \mid W^{(1)} \in \mathbf{W}^{(1)}, R_W \in \mathrm{pa}_{\mathcal{G}}(R_Y)\}$. By the assumed conditional independence properties, we have that

$$p(\mathbf{Z}^{(1)}, R_Y, \mathrm{pa}_{\mathcal{G}}(R_Y) \setminus \mathbf{Z}^{(1)})\Big|_{R_{\mathbf{D}}=1}$$

$$= \sum_{\mathbf{W}^{(1)}} p(\mathbf{Z}^{(1)}, \mathbf{W}^{(1)}, R_Y, \mathrm{pa}_{\mathcal{G}}(R_Y) \setminus \mathbf{Z}^{(1)})\Big|_{R_{\mathbf{D}}=1}$$

$$= \sum_{\mathbf{W}^{(1)}} p(\mathbf{Z}^{(1)} \mid \mathbf{W}^{(1)}, R_Y, \mathrm{pa}_{\mathcal{G}}(R_Y) \setminus \mathbf{Z}^{(1)}) p(\mathbf{W}^{(1)} \mid R_Y, \mathrm{pa}_{\mathcal{G}}(R_Y) \setminus \mathbf{Z}^{(1)})$$

$$\times p(R_Y, \mathrm{pa}_{\mathcal{G}}(R_Y) \setminus \mathbf{Z}^{(1)})\Big|_{R_{\mathbf{D}}=1}$$

$$= \sum_{\mathbf{W}^{(1)}} p(\mathbf{Z}^{(1)} \mid \mathbf{W}^{(1)}, R_{\mathbf{Z}}, R_{\mathbf{W}}, R_Y, \mathrm{pa}_{\mathcal{G}}(R_Y) \setminus (\mathbf{Z}^{(1)} \cup R_{\mathbf{W}}))$$

$$\times p(\mathbf{W}^{(1)} \mid R_{\mathbf{W}}, R_Y, \mathrm{pa}_{\mathcal{G}}(R_Y) \setminus (\mathbf{Z}^{(1)} \cup R_{\mathbf{W}})) p(R_Y, \mathrm{pa}_{\mathcal{G}}(R_Y) \setminus \mathbf{Z}^{(1)})\Big|_{R_{\mathbf{C}}=1, R_{\mathbf{Z}}=1, R_{\mathbf{W}}=1}$$

thus the claim follows by using the same reasoning again as in the proof of Theorem 3. $\qquad\square$

## B  Proofs for Section 6 (Identifiability Lost under Monotonicity)

**Theorem 6.** *Let $\mathcal{G}$ be an m-DAG that contains the edge $X_k^{(1)} \to R_{X_1}$ and the edges $R_{X_{j-1}} \to R_{X_j}$ for all $j = 2, \ldots, k$ (a self-censoring path). If $R_{X_{j-1}} \geq R_{X_j}$ for all $j = 2, \ldots, k$, then $p(X_k^{(1)})$ is not identifiable.*

*Proof.* Without loss of generality, we may assume that $\mathbf{O} = \emptyset$. If $k = 1$ the claim is immediate due to the self-censoring edge $X_1 \to R_{X_1}$. Suppose now that $k > 1$. We construct two models (parametrizations) such that the observed data laws agree but the full laws disagree between the models. We denote $\mathbf{Z} = \{X_k^{(1)}, R_{X_1}, \ldots, R_{X_k}\}$ and let $\mathbf{W} = (\mathbf{X}^{(1)} \cup \mathbf{R}) \setminus \mathbf{Z}$. We let the effects of variables in $\mathbf{W}$ on those in $\mathbf{Z}$ be null effects and vice versa for all value assignments. Further, we assume that following probability is constant and the same in both models

$$\prod_{V_i \in \mathbf{W}} p(V_i \mid \mathrm{pa}_{\mathcal{G}}(V_i)) = \alpha.$$

Next, we define the parametrizations along the self-censoring path in both models (we assume that $X_k^{(1)}$ is binary):

$$\beta_{0,1} = p_1(X_k^{(1)} = 0 \mid \mathrm{pa}_{\mathcal{G}}(X_k^{(1)})) = \gamma,$$
$$\beta_{0,2} = p_2(X_k^{(1)} = 0 \mid \mathrm{pa}_{\mathcal{G}}(X_k^{(1)})) = 1 - \gamma,$$
$$\beta_{1,0,1} = p_1(R_{X_1} = 0 \mid \mathrm{pa}_{\mathcal{G}}(R_{X_1}))\big|_{X_k^{(1)}=0} = \gamma,$$
$$\beta_{1,0,2} = p_2(R_{X_1} = 0 \mid \mathrm{pa}_{\mathcal{G}}(R_{X_1}))\big|_{X_k^{(1)}=0} = 1 - \gamma,$$
$$\beta_{1,1,1} = p_1(R_{X_1} = 0 \mid \mathrm{pa}_{\mathcal{G}}(R_{X_1}))\big|_{X_k^{(1)}=1} = 1 - \gamma,$$
$$\beta_{1,1,2} = p_2(R_{X_1} = 0 \mid \mathrm{pa}_{\mathcal{G}}(R_{X_1}))\big|_{X_k^{(1)}=1} = \gamma,$$

where $\gamma$ can be chosen freely as long as $\gamma \neq 0.5$, and

$$\beta_{2,0,i} = p_1(R_{X_2} = 0 \mid \mathrm{pa}_{\mathcal{G}}(R_{X_2}))\big|_{R_{X_1}=0} = 1,$$
$$\beta_{2,1,i} = p_2(R_{X_2} = 0 \mid \mathrm{pa}_{\mathcal{G}}(R_{X_2}))\big|_{R_{X_1}=1} = 0.5,$$
$$\vdots$$
$$\beta_{j,0,i} = p_1(R_{X_j} = 0 \mid \mathrm{pa}_{\mathcal{G}}(R_{X_k}))\big|_{R_{X_{k-1}}=0} = 1,$$
$$\beta_{j,1,i} = p_2(R_{X_j} = 0 \mid \mathrm{pa}_{\mathcal{G}}(R_{X_k}))\big|_{R_{X_{k-1}}=1} = 0.5,$$

for $i = 1, 2$. First, we must ensure that the observed data laws agree. To simplify the exposition, we define the following functions for the models:

$$f_i(x_k^{(1)}, r_{X_1}, \ldots, r_{X_k})$$
$$= \prod_{V_i \in \mathbf{Z}} p_i(V_i \mid \mathrm{pa}_{\mathcal{G}}(V_i))\bigg|_{X_k^{(1)}=x_k^{(1)}, R_{X_1}=r_{X_1}, \ldots, R_{X_k}=r_{X_k}}$$
$$= \beta_{0,i}^{1-x_k^{(1)}} (1 - \beta_{0,i})^{x_k^{(1)}} \beta_{1,x_k^{(1)},i}^{1-r_{X_1}} (1 - \beta_{1,x_k^{(1)},i})^{r_{X_1}} \prod_{j=2}^{k} \beta_{k,r_{X_{j-1}},i}^{1-r_{X_j}} (1 - \beta_{k,r_{X_{j-1}},i})^{r_{X_j}}.$$

We can simplify these functions by writing them separately for the two models

$$f_1(x_k^{(1)}, r_{X_1}, \ldots, r_{X_k}) = \frac{1}{2^{c(r_2,\ldots,r_k)}} \begin{cases} \gamma^2 & x_k^{(1)} = 0, r_{X_1} = 0, \\ (1-\gamma)^2 & x_k^{(1)} = 1, r_{X_1} = 0, \\ \gamma(1-\gamma) & r_{X_1} = 1, r_{X_j} \geq r_{X_{j+1}}, j = 1, \ldots, k-1 \\ 0 & \text{otherwise} \end{cases}$$

$$f_2(x_k^{(1)}, r_{X_1}, \ldots, r_{X_j}) = \frac{1}{2^{c(r_2,\ldots,r_k)}} \begin{cases} \gamma^2 & x_k^{(1)} = 1, r_{X_1} = 0, \\ (1-\gamma)^2 & x_k^{(1)} = 0, r_{X_1} = 0, \\ \gamma(1-\gamma) & r_{X_1} = 1, r_{X_j} \geq r_{X_{j+1}}, j = 1, \ldots, k-1 \\ 0 & \text{otherwise} \end{cases}$$

where $c(r_2, \ldots, r_k) = \sum_{j=2}^{k} r_{X_j}$. Denote $f_i^* = 2^{c(r_2,\ldots,r_k)} f_i$. We can now express the full law as follows for both models

$$p_i(\mathbf{X}^{(1)} = \mathbf{x}, \mathbf{R} = \mathbf{r}) = 2^{-c(r_{X_2},\ldots,r_{X_k})} \alpha f_i^*(x_k^{(1)}, r_{X_1}, \ldots, r_{X_k}).$$

We have the following system of equations for the observed data law, where the following functions must have the same value for $i = 1$ and $i = 2$ for all value assignments $\mathbf{X} = \mathbf{x}$.

$$h_i(\mathbf{x})$$
$$= p_i(\mathbf{X} = \mathbf{x}, \mathbf{R} = r(\mathbf{x}))$$
$$= \sum_{(\mathbf{x}'^{(1)}, \mathbf{r}') \text{ s.t. } (x_j'^{(1)}, r_{X_j}') \in \begin{cases} (x_j, 1) & \text{if } X_j \neq \mathrm{NA} \\ \{(0,0), (1,0)\} & \text{if } X_j = \mathrm{NA} \end{cases}} p_i(\mathbf{X}^{(1)} = \mathbf{x}', \mathbf{R} = \mathbf{r}').$$

Because the factors $\alpha$ and $c(\cdot)$ agree between the models, it suffices to focus on the functions $f_i^*$. If $X_k \neq \mathrm{NA}$, then it must be that $X_1 \neq \mathrm{NA}$ also, and it is easy to see that the value of the functions $h_i$ agree as $f_1^*(\cdot) = f_2^*(\cdot) = \gamma(1-\gamma)$ (or equal to zero) for all value assignments. When $X_j = \mathrm{NA}$ and $X_1 \neq \mathrm{NA}$, we have again $f_1^*(\cdot) = f_2^*(\cdot) = \gamma(1-\gamma)$ (or equal to zero) for all value assignments, thus all terms in the sum agree between the models. When $X_k = \mathrm{NA}$ and $X_1 = \mathrm{NA}$ the terms in the sum can be split into pairs such that each pair has a term with the factor $\gamma^2$ and a term with the factor $(1-\gamma)^2$ (corresponding to either $X_k = 0$

or $X_k = 1$ depending on the model). Thus the expression of $h_i$ is symmetric with respect to $\gamma$ and $(1 - \gamma)$. Thus $h_1(\mathbf{x}) = h_2(\mathbf{x})$ for all value assignments.

Next, we consider the marginal distribution of $X_k^{(1)}$ in both models

$$p_i(X_k^{(1)} = x_k) = \sum_{\mathbf{x} \setminus \{x_k^{(1)}\}} \sum_{\mathbf{r}} p_i(\mathbf{X} = \mathbf{x}, \mathbf{R} = \mathbf{r})$$

$$= \sum_{\mathbf{x} \setminus \{x_k^{(1)}\}} \sum_{\mathbf{r}} 2^{-c(r_{X_2}, \ldots, r_{X_k})} \alpha f_i^*(x_k^{(1)}, r_{X_1}, \ldots, r_{X_k})$$

We may again ignore the factors $\alpha$ and $c(\cdot)$ and focus on the following expression

$$\sum_{\mathbf{x} \setminus \{x_k^{(1)}\}} \sum_{\mathbf{r}} f_i^*(x_k^{(1)}, r_{X_1}, \ldots, r_{X_k})$$

Without loss of generality, we assume that the number of partially observed variables is $k$ and that all other partially observed variables are also binary. When at least one of $R_{X_1}, \ldots, R_{X_k}$ is not zero, the functions $f_i^*$ always output the value 0 for value assignments that violate monotonicity and the value $\gamma(1 - \gamma)$ otherwise. When all of $R_{X_1}, \ldots, R_{X_k}$ are equal to 0, then $f_1(x_k^{(1)} = 0, \cdot) = \gamma^2$ and $f_2(x_k^{(1)} = 0, \cdot) = (1 - \gamma)^2$. There are $2^{k-1}$ ways to choose the values of $X_1^{(1)}, \ldots, X_{k-1}^{(1)}$ and $k$ value assignments to $\mathbf{R}$ that do not violate monotonicity where at least one partially observed variable is not missing. Thus we have that

$$\sum_{\mathbf{x} \setminus \{x_k^{(1)}\}} \sum_{\mathbf{r}} f_i^*(x_k^{(1)}, r_{X_1}, \ldots, r_{X_k})$$

$$= 2^{k-1} \sum_{r_{X_1}, \ldots, r_{X_k}} f_i^*(x_k^{(1)}, r_{X_1}, \ldots, r_{X_k})$$

$$= 2^{k-1} \begin{cases} \gamma^2 + k\gamma(1 - \gamma) & i = 1, x_k^{(1)} = 0, \\ (1 - \gamma)^2 + k\gamma(1 - \gamma) & i = 1, x_k^{(1)} = 1, \\ (1 - \gamma)^2 + k\gamma(1 - \gamma) & i = 2, x_k^{(1)} = 0, \\ \gamma^2 + k\gamma(1 - \gamma) & i = 2, x_k^{(1)} = 1. \end{cases}$$

Because we can freely choose the value of $\gamma$ (as long as $\gamma \neq 0$), there are infinitely many models that agree on the observed data law, but disagree on the marginal distribution of $X_k^{(1)}$. $\qquad\square$

## C   Nonidentifiability construction for a bivariate self-censoring path

We consider a scenario with a locally monotone missingness mechanism with respect to $(R_X, R_Y)$ and a self-censoring path where there is also a true dependency between $X^{(1)}$ and $Y^{(1)}$. Thus we assume that $b \neq d$ and $c \neq e$ in the parametrization below.

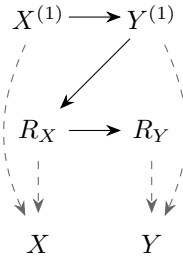

| $X^{(1)}$ | $p(X^{(1)})$ |
|---|---|
| 1 | $a$ |
| 0 | $1 - a$ |

| $X^{(1)}$ | $Y^{(1)}$ | $p(Y^{(1)}|X^{(1)})$ |
|---|---|---|
| 1 | 1 | $b$ |
| 1 | 0 | $1 - b$ |
| 0 | 1 | $d$ |
| 0 | 0 | $1 - d$ |

| $Y^{(1)}$ | $R_X$ | $p(R_X|Y^{(1)})$ |
|---|---|---|
| 1 | 1 | $c$ |
| 1 | 0 | $1 - c$ |
| 0 | 1 | $e$ |
| 0 | 0 | $1 - e$ |

| $R_X$ | $R_Y$ | $p(R_Y|R_X)$ |
|---|---|---|
| 1 | 1 | $f$ |
| 1 | 0 | $1 - f$ |
| 0 | 1 | $0$ |
| 0 | 0 | $1$ |

| $R_X$ | $R_Y$ | $X^{(1)}$ | $Y^{(1)}$ | $p(X^{(1)}, Y^{(1)}, R_X, R_Y)$ | $X$ | $Y$ | $p(X, Y, R_X, R_Y)$ |
|---|---|---|---|---|---|---|---|
| | | 1 | 1 | $abcf$ | 1 | 1 | $abcf = p_{11}$ |
| | | 1 | 0 | $a(1-b)ef$ | 1 | 0 | $a(1-b)ef = p_{10}$ |
| 1 | 1 | 0 | 1 | $(1-a)dcf$ | 0 | 1 | $(1-a)dcf = p_{01}$ |
| | | 0 | 0 | $(1-a)(1-d)ef$ | 0 | 0 | $(1-a)(1-d)ef = p_{00}$ |
| | | 1 | 1 | $abc(1-f)$ | 0 | | $a[bc + (1-b)e](1-f) = p_{0\mathrm{NA}}$ |
| | | 1 | 0 | $a(1-b)e(1-f)$ | | NA | |
| 1 | 0 | 0 | 1 | $(1-a)dc(1-f)$ | | | |
| | | 0 | 0 | $(1-a)(1-d)e(1-f)$ | 1 | | $(1-a)[dc+(1-d)e](1-f) = p_{1\mathrm{NA}}$ |
| | | 1 | 1 | 0 | | 0 | 0 |
| | | 1 | 0 | 0 | | | |
| 0 | 1 | 0 | 1 | 0 | NA | | |
| | | 0 | 0 | 0 | | 1 | 0 |
| | | 1 | 1 | $ab(1-c)$ | | | $(1-a)[(1-d)(1-e) + d(1-c)]$ |
| | | 1 | 0 | $a(1-b)(1-e)$ | NA | NA | $+ a[(1-b)(1-e) + b(1-c)]$ |
| 0 | 0 | 0 | 1 | $(1-a)d(1-c)$ | | | |
| | | 0 | 0 | $(1-a)(1-d)(1-e)$ | | | $= p_{\mathrm{NANA}}$ |

The probabilities $p_{11}$, $p_{10}$, $p_{01}$, $p_{00}$, $p_{0\mathrm{NA}} + p_{1\mathrm{NA}}$, and $p_{\mathrm{NANA}}$ are observed and they sum to 1. Note that $p_{0\mathrm{NA}}$ and $p_{1\mathrm{NA}}$ cannot be specified separately because they are functionally dependent on other probabilities. We use shortcut notations $\gamma_1 = p_{11}/p_{01}$ and $\gamma_0 = p_{10}/p_{00}$. When parameter $a$ is fixed, the other parameters can be solved from the observed probabilities as follows:

$$b = \frac{\gamma_0 - \frac{a}{1-a}\gamma_1\gamma_0}{\gamma_0 - \gamma_1},$$

$$c = \frac{p_{11}(\gamma_0 - \gamma_1)}{(1-a)\gamma_0\gamma_1 - a\gamma_1},$$

$$d = \frac{\gamma_0 - \frac{a}{1-a}}{\gamma_0 - \gamma_1},$$

$$e = \frac{p_{10}(\gamma_0 - \gamma_1)}{a\gamma_0 - (1-a)\gamma_0\gamma_1},$$

$$f = \frac{p_{11} + p_{10} + p_{01} + p_{00}}{p_{11} + p_{10} + p_{01} + p_{00} + p_{1\mathrm{NA}} + p_{0\mathrm{NA}}}.$$

It remains to check that the solutions for $b$, $c$, $d$, $e$, $f$ are between 0 and 1. This directly holds for parameter $f$ that does not depend on parameter $a$. The constraints $0 < b < 1$ and $0 < d < 1$ induce a necessary condition $\min(\gamma_0, \gamma_1) < a/(1-a) < \max(\gamma_0, \gamma_1)$.

In order to present a specific construction for nonidentifiability, consider the observed probabilities $p_{11} = 1/5$, $p_{10} = 1/10$, $p_{01} = 1/10$, $p_{00} = 1/5$, $p_{1\mathrm{NA}} = 1/20$ and $p_{0\mathrm{NA}} = 1/10$. We obtain $\gamma_1 = 2$ and $\gamma_0 = 1/2$, which implies $1/3 < a < 2/3$. Investigation of the constraints $0 < c < 1$ and $0 < e < 1$ reduces this interval to $9/20 < a < 11/20$. This means that we may define two models by picking two different values of $a$ from this interval and then solve the values of the other parameters. The two models will agree on all observed distributions but differ by unobserved distributions. As an illustration consider the models $\mathcal{M}_1$ with $a = 7/15 \approx 0.47$ and $\mathcal{M}_2$ with $a = 8/15 \approx 0.53$.

| Parameter | $\mathcal{M}_1$ | $\mathcal{M}_2$ |
|---|---|---|
| $a$ | 7/15 | 8/15 |
| $b$ | 12/21 | 3/4 |
| $c$ | 15/16 | 5/8 |
| $d$ | 1/4 | 9/21 |
| $e$ | 5/8 | 15/16 |
| $f$ | 4/5 | 4/5 |

The table below shows the probabilities of the full law under $\mathcal{M}_1$ and $\mathcal{M}_2$. It can be seen that the distributions differ when $R_X = 0$ and $R_Y = 0$ and are otherwise the same. This means the full laws differ between the models $\mathcal{M}_1$ and $\mathcal{M}_2$ while the observed data law are the same. By summing the probabilities over $R_X$ and $R_Y$ it can be confirmed that also the target laws and the marginal distributions $P(X^{(1)})$ and $P(Y^{(1)})$ differ between the models $\mathcal{M}_1$ and $\mathcal{M}_2$.

| $R_X$ | $R_Y$ | $X^{(1)}$ | $Y^{(1)}$ | $p(X^{(1)}, Y^{(1)}, R_X, R_Y)$ | $\mathcal{M}_1$ | $\mathcal{M}_2$ |
|---|---|---|---|---|---|---|
| | | 1 | 1 | $abcf$ | $\frac{1}{5}$ | $\frac{1}{5}$ |
| 1 | 1 | 1 | 0 | $a(1-b)ef$ | $\frac{1}{10}$ | $\frac{1}{10}$ |
| | | 0 | 1 | $(1-a)dcf$ | $\frac{1}{10}$ | $\frac{1}{10}$ |
| | | 0 | 0 | $(1-a)(1-d)ef$ | $\frac{1}{5}$ | $\frac{1}{5}$ |
| | | 1 | 1 | $abc(1-f)$ | $\frac{1}{20}$ | $\frac{1}{20}$ |
| 1 | 0 | 1 | 0 | $a(1-b)e(1-f)$ | $\frac{1}{40}$ | $\frac{1}{40}$ |
| | | 0 | 1 | $(1-a)dc(1-f)$ | $\frac{1}{40}$ | $\frac{1}{40}$ |
| | | 0 | 0 | $(1-a)(1-d)e(1-f)$ | $\frac{1}{20}$ | $\frac{1}{20}$ |
| | | 1 | 1 | 0 | 0 | 0 |
| 0 | 1 | 1 | 0 | 0 | 0 | 0 |
| | | 0 | 1 | 0 | 0 | 0 |
| | | 0 | 0 | 0 | 0 | 0 |
| | | 1 | 1 | $ab(1-c)$ | $\frac{1}{60}$ | $\frac{3}{20}$ |
| 0 | 0 | 1 | 0 | $a(1-b)(1-e)$ | $\frac{3}{40}$ | $\frac{1}{120}$ |
| | | 0 | 1 | $(1-a)d(1-c)$ | $\frac{1}{120}$ | $\frac{3}{40}$ |
| | | 0 | 0 | $(1-a)(1-d)(1-e)$ | $\frac{3}{20}$ | $\frac{1}{60}$ |

