# OpenReview forum: "Monotone Missing Data: A Blessing and a Curse"
_TMLR — Accepted by TMLR_

### Review · Reviewer_x5Ds · 2025-03-06

**Summary Of Contributions:**

The authors consider identification of monotone missingness mechanism in graphical models. They show that in some graphs identifiability can be gained compared to the non-monotone case, while, somewhat surprisingly, there can also be a loss of identifiability.

**Audience:**

Yes

**Claims And Evidence:**

Yes

**Requested Changes:**

- This is just a suggestion, but it might make sense to take the two nice examples used in the beginnings of Section 4 and 5 and to put them in a separate motivation section before Sections 4 and 5.
- The notation p(X,O,R) > 0 and the surrounding discussion on page 4 is very confusing to me (though again this seems to be something coming out of this specific literature). First, what does "p(X,O,R) > 0" mean? (with random variables). Second, it is clear that adding the assumption of monotonicity may induce (extreme) dependence, which simply reduces the support of (X,O,R), but how exactly is this a problem for positivity? My understanding is that this comes later on page 5, where it is explained why an earlier result cannot be used and has to do with the fact that we are interested in R_k=1 specifically. However, this all might be explained better.
- Related to this in the example at the end of page 10/beginning of page 11, can you explain (ideally in the example of cognitivie abilities), how the monotonicity exactly breaks identifiability?
- The notation of min R_C seems to be somewhat hidden at the end of page 7. As this seems quite important, it might make sense to put it somewhere more prominently, or at least define it before the result.
- End of page 6: There is an "also" too much

**Strengths And Weaknesses:**

Strengths
------------
- The paper is well written and motivated (though even with that, the paper is still a dry read)
- Monotone missingness is important and discussing the identifiability possible in this case is important
- It is quite surprising that some cases are actually not identifiable when monotonicity is present!

Weaknesses
-------------------
- The paper is hard to read for somebody who is not familiar with the literature on identification in graphical misisng data models. Unfortunately I do not see a clear fix for this.

---

> ### Author Response · Authors · 2025-09-22
>
> ## Weaknesses
>
> * We agree that the paper might be challenging for readers unfamiliar with graphical missing data models and identification theory. We will carefully review the text for opportunities to improve the presentation in this regard. We also hope that the proposed changes listed below will help alleviate this issue. We are open to any suggestions to make the paper more accessible to a broader audience.
>
> ## Requested Changes
>
> * By $p(X, O, R) > 0$, we simply mean that for all value assignments (X = x, O = o, R = r), the joint distribution has a positive probability, i.e., $p(X = x, O = o, R = r) > 0$. We will make this explicit in the text. We agree that an alternative way to describe monotonicity is the corresponding reduction in the support of $(X, O, R)$, but the assumptions in the related relevant literature on identification always include a positivity assumption for the distribution of $(X, O, R)$ (either explicitly or implicitly). Thus it is in our opinion more intuitive to cast the issue of monotonicity as an assumption on non-positivity for specific value assignments of $(X, O, R)$ where monotonicity is violated. The reviewer is correct in stating that the main issue comes from trying to apply existing methods and results under monotonicity (page 5). We will further clarify the discussion on this topic in the paper.
>
> * Regarding the example on cognitive abilities (pages 10--11), we will add an explanation on why monotonicity renders the full law non-identifiable in this example.
>
> * We will move the explanation for the notation $\min R_C$ before Theorem 1 where it is used for the first time, and will revise it as follows: "For convenience, we use the notation $\min R_C$ to denote the random variable that takes the smallest value among members of $R_C$. In other words, $\min R_C$ is 0 if at least one response indicator in $R_C$ is 0 and otherwise its value is 1, i.e., when all response indicators in $R_C$ have the value 1."
>
> * We will replace the extra "also" with "from" at the end of page 6, so that the sentence is correct.

---

> > ### Comment · Reviewer_x5Ds · 2025-10-22
> > **Thank you for addressing the comments**
> >
> > I feel my comments have been adressed adequately

---

### Review · Reviewer_31LF · 2025-09-20

**Summary Of Contributions:**

- The authors propose identification and non-identification results for graphical models of missing data with monotone missing data. These restrictions are relatively understudied in the current graphical modeling literature on missing data. The author's identification results are interesting and, to my knowledge, novel
- There is a proposed connection between the identifiability result and imputation. This connection, at least in the current presentation of the paper, is much less clear to me and perhaps underdeveloped

**Audience:**

Yes

**Broader Impact Concerns:**

No concerns

**Claims And Evidence:**

Yes

**Requested Changes:**

Please see the Weakness section above --- I'd recommend making some connections to D-separation and addressing the comments regarding connections to imputation---maybe it's too ambitious to have a completely general procedure, but it would be good to be explicit about the procedure for at least some of the graphs/class of graphs discussed here

Minor typos:

-  Furthermore, we say that a such a value --> remove the first "a"

- "Second, there is no sound and complete identifiability algorithm for nonmonotone missingness which could serve as a starting point, as the result by Nabi et al. (2020) relies on the OR factorization in (4) which does not directly translate to monotone settings". This sentence is a little unclear. I think you mean to say there is no sound and complete result for nonmonotone missingness with functional dependence?


Edit:

Acknowledging that my comments have been sufficiently addressed and updating my review accordingly.

**Strengths And Weaknesses:**

Strengths:
- Positivity constraints in missing data are understudied and the authors present interesting (and to my knowledge, novel)  identification and non-identification results for monotone missingness

Weaknesses:

- The authors point out that d-separation is not sufficient to determine all independence/dependence relations when there is functional determinism in the graph due to monotonicity. However, the authors do not make sufficiently clear whether D-separation (uppercase D rather than lowercase) is also not suitable for this purpose --- https://ftp.cs.ucla.edu/pub/stat_ser/r116.pdf. If it is suitable, it would be great to present the results in terms of D-separation statements (and perhaps simplify and strengthen the existing results presented in the paper). In particular, it would be interesting to draw connections to this paper --- Identifying Causal Effects Under Functional Dependencies -- Chen and Darwiche (https://www.mdpi.com/1099-4300/26/12/1061). It presents extensions of the do-calculus to account for functional determinism and uses D-separation for this purpose. Their results might apply to monotone missingness as well

- The extent to which the identification results easily generalize to imputation is unclear; this contribution feels ad hoc and underdeveloped in the current version of the paper. In particular, the statements of the form "The identifiability of the full law in the case of Figure 1 implies that multiple imputation can be used to estimate the target law p(X(1), Y (1))" are not sufficiently rigorous. Identification alone is not sufficient---an imputation procedure that accounts for the manner in which the model is identified is also important. E.g., Applying MICE, which assume the data are MAR, probably would not directly work for models that are MNAR---if that is the implication being made, this should be supported with theory supporting why this is the case. If the authors mean that a sequential imputation procedure like the one in Karvanen and Tikka (2024) might be applied to certain identified graphs that have a sequential identification structure, this is also not sufficiently clear. In my understanding, that procedure assumed the absence of colluders, so would still need some modification when they are present. Other MNAR imputation procedures for graphical models of missing data, such as [Ren et al (2021)](https://journals.sagepub.com/doi/pdf/10.1177/09622802231188520?casa_token=O1x-eLPYMM8AAAAA:pg_49g_YG87H1Cd66C4ir4SLzupdkFEKxSlj1X64JCFCa6slosApFOZ6P8q1Im860hzgDPtHLl3ivK4) and [Phung et al (2025)](https://www.arxiv.org/pdf/2507.16107), also assume no colluders.

To summarize the above, I agree with the claim that imputation is possible when the model is fully identified, but the authors do not make it sufficiently rigorous/clear, what the imputation procedure should be. In my opinion, this detracts from the genuinely interesting identification results presented in the paper.

---

> ### Author Response · Authors · 2025-09-22
>
> ## Weaknesses
>
> * We agree that discussing D-separation would be appropriate. We note that in our setting the dependencies between the response indicators are not true functional dependencies in the sense that the response indicators would always be deterministic; this holds only for some value assignments of the response indicators. Thus using D-separation may also lead to incorrect conclusions and we cannot apply functional elimination as Chen and Darwiche (2024) do. We will modify the text to discuss these points further.
>
> * We strongly agree that the connections to multiple imputation are underdeveloped and may detract from the new identifiability and non-identifiability results that are the main focus of the paper. Given the aforementioned points and the fact that multiple imputation is discussed only briefly in the main text, we suggest that we completely omit multiple imputation from the paper. We feel that this subject would be better explored in a separate paper.
>
> ## Requested Changes
>
> * As mentioned before, we will extend the text regarding D-separation and functional dependence. We propose that multiple imputation is omitted from the paper.
>
> * We will correct the typo, and the clarify the sentence regarding Nabi et al. (2020) in the Discussion. Our intention was to simply state that their result cannot be used nor extended directly to the monotone setting due to the reliance on the OR factorization.

---

> > ### Comment · Reviewer_31LF · 2025-10-08
> > **Thank you, my comments have been addressed**
> >
> > Thank you for your response and revisions! I feel my comments have been well addressed

---

### Review · Reviewer_RXd2 · 2025-10-06

**Summary Of Contributions:**

The primary contribution is the analysis of identification under monotonicity. Interestingly, monotonicity can enable identification of distributions that are otherwise nonidentifiable under general (nonmonotone) missing data. Conversely, monotonicity can destroy identifiability for models that would otherwise be identifiable.

**Audience:**

Yes

**Claims And Evidence:**

Yes

**Requested Changes:**

The requested changes are noted above in the section on strengths and weaknesses. The key point is to ensure that acknowledgement and credit are given where they are due.

**Strengths And Weaknesses:**

I find the contributions of this work to be both novel and valuable. However, I have a few comments aimed primarily at improving the writing for greater clarity.

Abstract: The abstract would benefit from being framed in more general terms. Avoid using specific technical terminology such as “colluders,” which may not be familiar to a broad audience. Instead, emphasize the central idea of monotonicity—briefly explain what it means, why it is important, and how it motivates the proposed work. A short, intuitive explanation in plain language would make the abstract more engaging and accessible

The concept of monotonic missingness should be explained more clearly, ideally with a simple illustrative example involving three or four variables. Providing this early in the paper (e.g., on page 1) would help readers grasp the intuition behind the concept before delving into formal definitions.

Local monotone is mentioned but is there a notion of global monotone as well? May be worth stating somewhere.

Some terminology appears to deviate from established usage in the literature, which may confuse readers familiar with prior frameworks. For instance, the term missingness mechanism is used here to denote a distribution, whereas in Pearl’s framework it refers to the R variables. Similarly, the target law in this work corresponds to the joint distribution in Pearl’s terminology. It would be helpful to clearly state these distinctions to avoid ambiguity.

Definition 1 could be made more precise. In Rx-->Ry, here X is used as a variable, but previously X also denotes a set of proxies, which may cause confusion. Consider using W or another symbol instead. Explain what "<" indicates in the definition. In the last sentence of the definition, specify which assignment the phrase “such a value assignment” refers to.

In the paragraph following Definition 1, the authors mention two ways of graphical representation. Since only one is used consistently throughout the paper, it may suffice to describe just that one.

Again about notations, self-censoring is called as self-masking in Pearl's work. See https://www.ijcai.org/proceedings/2018/705 Acknowledging this alternative terminology would align the paper with existing literature.

When citing Nabi et al. for the non-identifiability of the full law, please also acknowledge that these results build upon earlier findings by Mohan et al. (2013) and Mohan & Pearl (2014). Specifically, Mohan et al. (2013) established the non-identifiability of the joint distribution under self-masking, and Mohan & Pearl (2014) showed non-identifiability when a path exists from Z to Rz where all intermediate nodes are colliders (later termed colluders).

Some of the cases discussed here have been examined in https://ftp.cs.ucla.edu/pub/stat_ser/mohan-ch34-acm-2021.pdf, although without taking monotonicity into account.

Overall, I find this work to be a valuable contribution and support its acceptance. I do not anticipate the need for further review from my side, unless the authors strongly disagree with any of the comments provided and wish to contest/discuss them.

---

> ### Author Response · Authors · 2025-10-09
>
> ## Requested changes
>
> * We have revised the abstract such that it uses more general terms regarding the graphical structures of non-identifiability.
>
> * We have added an illustrative example on monotonicity to the introduction on page 1 involving three variables.
>
> * Global monotonicity could be defined for example such that as each pair of response indicators connected by an edge are locally monotone, e.g., as in Figure 3(b). We do not mention this in the paper, as we only consider local instances of monotonicity.
>
> * When introducing the terminology, we now note that "missingness mechanism" sometimes refers to the $R$ variables, and similarly that the target law is sometimes called the "joint distribution". The use of the terms "target law" and "full law" is motivated by the ambiguity of "joint distribution", which could refer to the either one of the aforementioned joint distributions.
>
> * We have swapped $R_X$ to $R_Z$ and $R_Y$ to $R_W$ in Definition 1 and in the following paragraph. We also clarify that "such a value assignment" refers to the assignment of $r$ to $R$.
>
> * We note that both graphical representations of the assumed local monotonicity are used in the paper (mainly to avoid edges overlapping with the $\leq$ sign), e.g., in Figure 3(a) and Figure 3(b).
>
> * We now acknowledge that self-censoring is also referred to as self-masking.
>
> * We now mention Mohan et al. (2013) and Mohan et al. (2014) as seminal works leading up to the result by Nabi et al. When discussing graphical structures related to non-identifiability, we also mention the self-censoring result related to the target law by Mohan et al. (2013).
>
> * Regarding the identifiability example without monotonicity in Section 6, we now mention that this example has been considered before by Mohan (2022).

---

> > ### Comment · Reviewer_RXd2 · 2025-10-09
> >
> > Thank you!
> > Finally, although not strictly necessary, it might be helpful to include an example after each of the new results (theorems) presented here, provided there is sufficient space. Imagine a student reading the paper for the first or second time: the theorems might initially seem difficult to follow, but an accompanying example would make them much easier to understand.

---

> > > ### Author Response · Authors · 2025-10-10
> > >
> > > Thank you for the suggestion! We agree, but this is already the case in the paper (with the exception of Theorem 1, which is much simpler than the other theorems). Each Theorem from 2 to 5 is followed by an example (2 examples for Theorem 5), which map directly to the m-DAGs in Figure 2. Similarly for Theorem 6, examples are given in the preceding paragraph related to the m-DAGs of Figure 3.

---

### Decision · Action_Editor_rR1y · 2025-11-04

**Recommendation:** Accept as is

**Additional Comments:**

The authors have addressed all reviewers' concerns.

**Audience:**

Yes

**Audience Explanation:**

All reviewers agree that the manuscript has an audience. The topic is relevant to the causality subcommunity and the TMLR readership. The audience might be relatively smaller as compared to the whole machine learning community, but it is important to let the ML community know of these topics and developments, and to increase the diversity of topics being studied.

**Claims And Evidence:**

Yes

**Claims Explanation:**

All reviewers agree that the manuscript presents appropriate evidence for its claims. Identifiability (of joint distribution variables and response indicators) with non-monotone missingness has been well studied. This manuscript studies monotone missingness and show that monotonicity enables identifiability that would otherwise be nonidentifiable, and in addition, monotonicity also renders otherwise identifiable distributions nonidentifiable.